# Set-Based Training for Neural Network Verification

**Lukas Koller**  *lukas.koller@tum.de*
*Technical University of Munich*

**Tobias Ladner**  *tobias.ladner@tum.de*
*Technical University of Munich*

**Matthias Althoff**  *althoff@tum.de*
*Professorship for Cyber-Physical Systems*
*Technical University of Munich*

**Reviewed on OpenReview:** *https://openreview.net/forum?id=nOlzHrAWIA*

## Abstract

Neural networks are vulnerable to adversarial attacks, i.e., small input perturbations can significantly affect the outputs of a neural network. Therefore, to ensure safety of neural networks in safety-critical environments, the robustness of a neural network must be formally verified against input perturbations, e.g., from noisy sensors. To improve the robustness of neural networks and thus simplify the formal verification, we present a novel set-based training procedure in which we compute the set of possible outputs given the set of possible inputs and compute for the first time a gradient set, i.e., each possible output has a different gradient. Therefore, we can directly reduce the size of the output enclosure by choosing gradients toward its center. Small output enclosures increase the robustness of a neural network and, at the same time, simplify its formal verification. The latter benefit is due to the fact that a larger size of propagated sets increases the conservatism of most verification methods. Our extensive evaluation demonstrates that set-based training produces robust neural networks with competitive performance, which can be verified using fast (polynomial-time) verification algorithms due to the reduced output set.

## 1 Introduction

Neural networks demonstrate impressive performance for complex tasks, such as speech recognition (Hinton et al., 2012) or object detection (Wang et al., 2023). However, many neural networks are sensitive to input perturbations (Szegedy et al., 2014): Small, carefully chosen input perturbations can lead to unexpected outputs. This behavior is problematic for the adoption of neural networks in safety-critical environments, where the input often contains noisy sensor data or is subject to external disturbances, e.g., autonomous vehicle control (Ye et al., 2022) or airborne collision avoidance (Irfan et al., 2020). Thus, the formal verification of neural networks gained interest in recent years (Brix et al., 2023; 2024). Given a set of inputs, the goal of formal verification of neural networks is to find a mathematical proof that the neural network returns the correct output for every input from the set. We focus on training and verifying robust neural networks. For the robustness verification, the goal is to verify that within bounded input perturbations, which are typically modeled as $\ell_\infty$-balls of radius $\epsilon \in \mathbb{R}_{>0}$, there is no adversarial input that is classified incorrectly.

In this work, we address the challenge of training and verifying the robustness of a neural network with a novel set-based training procedure. Adversarially trained neural networks (Madry et al., 2018; Zhang et al., 2019) achieve great empirical robustness (against adversarial attacks). However, they remain hard to formally verify due to large approximation errors, which increase the size of enclosures of their output

**Single Gradient**

Figure 1: Comparing our gradient set with the gradient of other robust training approaches for the same neural network in the output space. The blue area shows the exact output set of the neural network. Other training approaches ((a) IBP (Gowal et al., 2019), (b) DiffAi (Mirman et al., 2018) and (c) SABR (Müller et al., 2023)) propagate intervals to compute a single gradient. On the contrary, our set-based training (d) computes a gradient set based on the position (for accuracy) and size (for robustness) of the output set.

sets. Thus, we propose a novel set-based training that can explicitly enforce small output sets to simplify a subsequent formal verification. During training, we enclose the output set of the neural network and compute a gradient set, i.e., each possible output has a different gradient. We can directly enforce smaller output enclosures by choosing gradients that point toward the center of the enclosure. To compute the gradient set, we use a set-based loss function, which considers the position (for accuracy) and the size of the output enclosure (for robustness). Previous works are limited to a single gradient for training, thereby discarding much set-based information. Fig. 1 compares our gradient set with the gradient of three robust training approaches: Interval bound propagation (IBP) (Gowal et al., 2019), DiffAi (Mirman et al., 2018) and SABR (Müller et al., 2023). IBP computes bounds of the output set and uses the gradient of the worst-case output; DiffAi uses a zonotope enclosure to compute tighter bounds compared to IBP; SABR uses the gradient of the worst case of a smaller input region to reduce regularization by large approximation errors. Subsequently, we provide an overview of related work.

**Formal Verification of Neural Networks** The formal verification of neural networks is computationally challenging, i.e., the problem is NP-hard with only rectified linear unit (ReLU) activation functions (Katz et al., 2017); thus, even verifying small neural networks can take a long time. Most formal verification approaches either (i) formulate an optimization problem, which is solved with an off-the-shelf solver, e.g., (mixed-integer) linear programming (Singh et al., 2019b; Zhang et al., 2022; Müller et al., 2022) or satisfiability modulo theories (Katz et al., 2017; Wu et al., 2024), or (ii) use reachability analysis with efficient set representations, e.g., zonotopes (Girard, 2005), and set-based computations to enclose the output set of a neural network (in polynomial time) (Gehr et al., 2018; Singh et al., 2019a; Wang et al., 2021; Kochdumper et al., 2023; Ladner & Althoff, 2023; Lemesle et al., 2024). If the enclosure of the output set is sufficiently

tight, it can be used to formally verify a neural network. The challenge for most reachability-based approaches is the enclosure of nonlinearities in a neural network, which limits their scalability because each nonlinearity adds an approximation error that accumulates for large neural networks. Therefore, in practice, often branch-and-bound procedures (Bunel et al., 2020; Wang et al., 2021; Durand et al., 2022; Ferrari et al., 2022) are used to recursively split the verification problem into smaller and simpler subproblems, e.g., by exploiting the piecewise linearity of the ReLU activation function. Due to the recursive process, branch-and-bound algorithms have exponential runtime in the worst case. We want to train robust neural networks that can be quickly verified without splitting the verification problem.

On the contrary, neural networks can be falsified by adversarial attacks, i.e., small input perturbations that lead to an incorrect output. Often, adversarial attacks are fast to compute and effective at provoking incorrect outputs (Goodfellow et al., 2015). The fast gradient sign method (FGSM) and projected gradient descent (PGD) are the most prominent approaches. The FGSM is a single-step gradient-based adversarial attack that efficiently generates adversarial attacks (Goodfellow et al., 2015). PGD uses multiple iterations of FGSM to compute stronger adversarial attacks (Kurakin et al., 2017). However, neural networks can be robust for one type of attack but can remain vulnerable to another type of attack (Madry et al., 2018); thus, it is necessary to formally verify the robustness of neural networks.

**Training Robust Neural Networks**  The training objective of a robust neural network is typically formulated as a min-max optimization problem (Madry et al., 2018): minimize the worst-case loss within a set of possible inputs. Computing the worst-case loss within a set is computationally difficult (Weng et al., 2018). Nonetheless, robust neural networks can be effectively trained by approximating the worst-case loss with adversarial attacks, e.g., computed using PGD (Madry et al., 2018).

The well-established tradeoff-loss (Zhang et al., 2019) combines a regular loss for accuracy with a boundary loss for robustness, which is approximated using adversarial attacks. The boundary loss pushes the decision boundary of a classifier away from the training samples and thereby improves the robustness of the trained neural network. However, neural networks trained with adversarial attacks remain hard to formally verify.

Therefore, some approaches combine the training and formal verification of neural networks. In these works, the approximation of a worst-case loss is replaced by an upper bound, guaranteeing that no perturbation will lead to an incorrect output. Different methods for computing an upper bound of the worst-case loss have been proposed: Interval bound propagation (IBP) (Gowal et al., 2019), linear relaxation (Zhang et al., 2020), (mixed-integer) linear programming (Wong & Kolter, 2018), or abstract interpretation (Mirman et al., 2018). IBP computes conservative output bounds and uses the worst case within the bounds for training and verification (Gowal et al., 2019) (Fig. 1a). However, large approximation errors can create an over-regularization and lead to poor performance (Müller et al., 2023).

Thus, by propagating smaller input sets, state-of-the-art robustness results are achieved (Müller et al., 2023) (Fig. 1b). However, a branch-and-bound algorithm with worst-case exponential-time complexity is used for their formal verification. Instead, we aim to train robust neural networks that can be quickly verified using polynomial-time verification algorithms. Therefore, we enclose the output set of a neural network using zonotopes, which is significantly tighter than using IBP. More closely related to our work is an approach using zonotopic enclosures during training (Mirman et al., 2018); however, much set-based information is discarded by only using the enclosure to bound the worst-case loss. All related approaches for training robust neural networks consider a single gradient. Conversely, our approach considers a gradient set (Fig. 1c), thereby simultaneously increasing the robustness of the neural network and simplifying the formal verification.

Other approaches combine IBP adversarial attacks computed in a latent space (Mao et al., 2023). Similarly, zonotopes can be partially propagated through a neural network to compute adversarial attacks in latent space (Balunović & Vechev, 2020); however, the partial propagation results in layer-wise training of the neural network, which has a significant computational overhead. Furthermore, the training of robust neural networks can be improved by using a specialized initialization for the parameters of a neural network (Shi et al., 2021).

**Contributions**  Our main contributions are:

- A novel set-based training procedure for robust neural networks that, for the first time, uses a gradient set for training that generalizes the well-established tradeoff-loss (Zhang et al., 2019). The gradient set enables direct control of the size of the output enclosure to improve the robustness of the neural network and simplify formal verification.

- A fast, batch-wise, and differentiable set-based forward propagation and backpropagation that is efficiently computed on a GPU. The set propagation uses analytical solutions for the image enclosure of typical nonlinear activation functions.

- An extensive empirical evaluation in which we demonstrate the competitive performance of our set-based training and compare it with state-of-the-art robust training approaches. Moreover, we include large-scale ablation studies to justify our design choices.

**Organization**  We introduce the required preliminaries in Sec. 2. In Sec. 3, we present our set-based training procedure, which benefits from a fast, batch-wise, and differentiable set propagation derived in Sec. 4. We provide an empirical evaluation, including ablation studies in Sec. 5. Finally, we conclude our findings in Sec. 6.

## 2 Preliminaries

### 2.1 Notation

Lowercase letters denote vectors and uppercase letters denote matrices. The $i$-th entry of a vector $x$ is denoted by $x_{(i)}$. For a matrix $A \in \mathbb{R}^{n \times m}$, $A_{(i,j)}$ denotes the entry in the $i$-th row and the $j$-th column, $A_{(i,\cdot)}$ denotes the $i$-th row, and $A_{(\cdot,j)}$ the $j$-th column. The identity matrix is written as $I_n \in \mathbb{R}^{n \times n}$. We use $\mathbf{0}$ and $\mathbf{1}$ to represent the vector or matrix (with appropriate size) that contains only zeros or ones. Given two matrices $A \in \mathbb{R}^{m \times n_1}$ and $B \in \mathbb{R}^{m \times n_2}$, their (horizontal) concatenation is denoted by $\begin{bmatrix} A & B \end{bmatrix} \in \mathbb{R}^{m \times (n_1+n_2)}$; if $n_1 = n_2$, their Hadamard product is the element-wise multiplication $(A \odot B)_{(i,j)} = A_{(i,j)} B_{(i,j)}$. The operation $\mathrm{Diag}(x) \in \mathbb{R}^{n \times n}$ returns a diagonal matrix with the entries of the vector $x \in \mathbb{R}^n$ on its diagonal. We denote sets with uppercase calligraphic letters. For a set $\mathcal{S} \subset \mathbb{R}^n$, we denote its projection to the $i$-th dimension by $\mathcal{S}_{(i)}$. Given two sets $\mathcal{S}_1 \subset \mathbb{R}^n$ and $\mathcal{S}_2 \subset \mathbb{R}^m$, we denote the Cartesian product by $\mathcal{S}_1 \times \mathcal{S}_2 = \{[s_1^\top \ s_2^\top]^\top \mid s_1 \in \mathcal{S}_1, s_2 \in \mathcal{S}_2\}$, and if $n = m$, we write the Minkowski sum as $\mathcal{S}_1 \oplus \mathcal{S}_2 = \{s_1 + s_2 \mid s_1 \in \mathcal{S}_1, s_2 \in \mathcal{S}_2\}$. For $n \in \mathbb{N}$, $[n] = \{1, 2, \ldots, n\}$ denotes the set of all natural numbers up to $n$. An $n$-dimensional interval $\mathcal{I} \subset \mathbb{R}^n$ with bounds $l, u \in \mathbb{R}^n$ is denoted by $\mathcal{I} = [l, u]$, where $\forall i \in [n]: l_{(i)} \leq u_{(i)}$. For a function $f: \mathbb{R}^n \to \mathbb{R}^m$, we abbreviate its evaluation for a set $\mathcal{S} \subset \mathbb{R}^n$ with $f(\mathcal{S}) = \{f(s) \mid s \in \mathcal{S}\}$. The derivative of a scalar function $f: \mathbb{R} \to \mathbb{R}$ is denoted as $f'(x) = \mathrm{d}/\mathrm{d}x \, f(x)$. Moreover, the gradient of a function $f: \mathbb{R}^n \to \mathbb{R}$ w.r.t. a vector $x \in \mathbb{R}^n$ is its element-wise derivative: $(\nabla_x f(x))_{(i)} = \partial/\partial x_{(i)} \, f(x)$, for $i \in [n]$. Analogously, we define the gradient of a function $f: \mathbb{R}^{n \times m} \to \mathbb{R}$ w.r.t. a matrix $A \in \mathbb{R}^{n \times m}$: $(\nabla_A f(A))_{(i,j)} = \partial/\partial A_{(i,j)} \, f(A)$, for $i \in [n]$ and $j \in [m]$.

### 2.2 Feed-Forward Neural Networks

A feed-forward neural network $\Phi_\theta: \mathbb{R}^{n_0} \to \mathbb{R}^{n_\kappa}$ consists of a sequence of $\kappa \in \mathbb{N}$ layers. For the $k$-th layer, $n_{k-1} \in \mathbb{N}$ denotes the number of input neurons and $n_k \in \mathbb{N}$ denotes the number of output neurons. A layer can either be a linear layer, which applies an affine map, or a nonlinear (activation) layer, which applies a nonlinear activation function element-wise.

**Definition 1** (Neural Network, (Bishop, 2006, Sec. 5.1)). *For an input $x \in \mathbb{R}^{n_0}$, the output $y = \Phi_\theta(x) \in \mathbb{R}^{n_\kappa}$ of a neural network $\Phi_\theta$ is computed by*

$$h_0 = x, \qquad\qquad h_k = L_k(h_{k-1}) \quad \text{for } k \in [\kappa], \qquad\qquad y = h_\kappa,$$

*where*

$$h_k = L_k(h_{k-1}) = \begin{cases} W_k \, h_{k-1} + b_k & \text{if } k\text{-th layer is linear,} \\ \phi_k(h_{k-1}) & \text{otherwise,} \end{cases}$$

*with weights $W_k \in \mathbb{R}^{n_k \times n_{k-1}}$, bias $b_k \in \mathbb{R}^{n_k}$, and nonlinear activation function $\phi_k$ which is applied element-wise.*

We denote the parameters of the neural network with $\theta$, which include all weight matrices and bias vectors from its linear layers.

**Training of Neural Networks**  We consider supervised training settings of a classification task, where a neural network is trained on a dataset $\mathcal{D} = \{(x_1, t_1), \ldots, (x_n, t_n)\}$, containing inputs $x_i \in \mathbb{R}^{n_0}$ with associated targets $t_i \in \{0, 1\}^{n_\kappa}$. A target $t_i \in \{0, 1\}^{n_\kappa}$ is a one-hot encoding of the target label $l_i \in [n_\kappa]$, i.e., $t_{i(j)} = 1 \iff j = l_i$ for all $j \in [n_\kappa]$. A loss function $L : \mathbb{R}^{n_\kappa} \times \mathbb{R}^{n_\kappa} \to \mathbb{R}$ measures how well a neural network predicts the targets. A typical loss function for classification tasks is the cross-entropy error:

$$L_{CE}(t, y) := -\sum_{i=1}^{n_\kappa} t_{(i)} \ln(p_{(i)}), \tag{1}$$

where ln denotes the natural logarithm and $p_{(i)} = \exp(y_{(i)})/(\exp(y)\,\mathbf{1})$ are the predicted class probabilities. The training goal of a neural network is to find parameters $\theta$ that minimize the total loss of the dataset $\mathcal{D}$ (Bishop, 2006, Sec. 5.2):

$$\min_\theta \sum_{(x_i, t_i) \in \mathcal{D}} L(t_i, \Phi_\theta(x_i)). \tag{2}$$

We revisit the training of a neural network to later augment it with sets. A popular algorithm to train a neural network is gradient descent (Bishop, 2006, Sec. 5.2.4): the parameters are randomly initialized (Glorot & Bengio, 2010) and iteratively optimized using the gradient of the loss function. We denote the gradient of the loss function $L$ w.r.t. the output of the $k$-th layer $h_k$ as:

$$g_k := \nabla_{h_k} L(t, y), \tag{3}$$

where $y = \Phi_\theta(x)$ for input $x \in \mathbb{R}^{n_0}$. The weight matrix $W_k$ and bias vector $b_k$ of the $k$-th layer are updated as (Bishop, 2006, Sec. 5.3)

$$W_k \leftarrow W_k - \eta \, \nabla_{W_k} L(t, y) = W_k - \eta \, g_k \, h_{k-1}^\top, \qquad b_k \leftarrow b_k - \eta \, \nabla_{b_k} L(t, y) = b_k - \eta \, g_k, \tag{4}$$

where $\eta \in \mathbb{R}_{>0}$ is the step size of gradient descent, i.e., the learning rate. The gradients $g_k$ are efficiently computed with backpropagation (Bishop, 2006, Sec. 5.3): by utilizing the chain rule, the gradient $g_\kappa$ of the last layer is propagated backward through all neural network layers.

**Proposition 1** (Backpropagation, (Bishop, 2006, Sec. 5.3))**.** *Let $y \in \mathbb{R}^{n_\kappa}$ be an output of a neural network with target $t \in \mathbb{R}^{n_\kappa}$. The gradients $g_k$ are computed in reverse order as*

$$g_\kappa = \nabla_y L(t, y), \qquad\qquad g_{k-1} = \begin{cases} W_k^\top \, g_k & \text{if } k\text{-th layer is linear,} \\ \mathrm{Diag}\big(\phi_k'(h_{k-1})\big) \, g_k & \text{otherwise,} \end{cases}$$

*for $k = \kappa, \kappa - 1, \ldots, 1$.*

From now on, we refer to this (standard) neural network training as point-based training.

## 2.3  Set-Based Computation

Our approach extends point-based training to sets, which we represent by zonotopes. A zonotope is a convex set representation describing the Minkowski sum of a finite number of line segments.

**Definition 2** (Zonotope, (Girard, 2005, Def. 1))**.** *Given a center $c \in \mathbb{R}^n$ and a generator matrix $G \in \mathbb{R}^{n \times q}$, a zonotope $\mathcal{Z} \subset \mathbb{R}^n$ is defined as*

$$\mathcal{Z} = \left\{ c + \sum_{i=1}^q G_{(i,\cdot)} \, \beta_{(i)} \ \middle|\ \beta \in [-\mathbf{1}, \mathbf{1}] \right\} =: \langle c, G \rangle_Z.$$

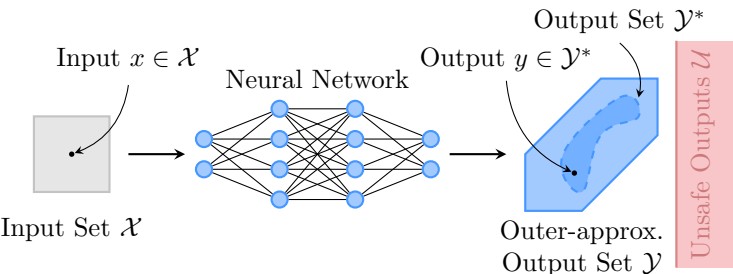

Figure 2: Verifying the local robustness of a neural network.

Subsequently, we define several operations for zonotopes used in our training approach. Please note that for the complexity analysis, we only consider the number of binary operations and neglect the computational effort of unary operations. Moreover, we consider the textbook method and do not assume any special numerical tricks that have been developed for large matrices.

**Proposition 2** (Interval Enclosure, (Althoff, 2010, Prop. 2.2))**.** *A zonotope $\mathcal{Z} = \langle c, G \rangle_Z$ with $c \in \mathbb{R}^n$ and $G \in \mathbb{R}^{n \times q}$ is enclosed by the interval $[l, u] \supseteq \mathcal{Z}$, where*

$$l = c - |G|\, \mathbf{1}, \qquad\qquad\qquad u = c + |G|\, \mathbf{1},$$

*and $|\cdot|$ computes the element-wise absolute value. Computing an interval enclosure is in $\mathcal{O}(n\,q)$.*

**Proposition 3** (Minkowski Sum, (Althoff, 2010, Prop. 2.1 and Sec. 2.4))**.** *The Minkowski sum of a zonotope $\mathcal{Z} = \langle c, G \rangle_Z$ and an interval $\mathcal{I} = [l, u] \subset \mathbb{R}^n$ with $c, l, u \in \mathbb{R}^n$ and $G \in \mathbb{R}^{n \times q}$ is computed as*

$$\mathcal{Z} \oplus \mathcal{I} = \left\langle c + \tfrac{1}{2}(u + l), \left[ G \ \tfrac{1}{2}\operatorname{Diag}(u - l) \right] \right\rangle_Z,$$

*and has time complexity $\mathcal{O}(n)$.*

**Proposition 4** (Affine Map, (Althoff, 2010, Sec. 2.4))**.** *The result of an affine map $f \colon \mathbb{R}^n \to \mathbb{R}^m$, $x \mapsto W\,x + b$ with $W \in \mathbb{R}^{m \times n}$ and $b \in \mathbb{R}^m$ applied to a zonotope $\mathcal{Z} = \langle c, G \rangle_Z$ with $c \in \mathbb{R}^n$ and $G \in \mathbb{R}^{n \times q}$ is*

$$f(\mathcal{Z}) = \{ f(z) \mid z \in \mathcal{Z} \} = W\,\mathcal{Z} \oplus b = \langle W\,c + b, W\,G \rangle_Z,$$

*and has time complexity $\mathcal{O}(m\,n\,q)$.*

During set-based training, we want to reduce the size of the output sets. However, determining the volume of a zonotope is computationally demanding (Elekes, 1986). Nevertheless, we can effectively approximate the size of a zonotope by its F-radius (Combastel, 2015): The F-radius of a zonotope is the Frobenius norm of its generator matrix.

**Proposition 5** (F-Radius, (Combastel, 2015, Def. 3))**.** *For a zonotope $\mathcal{Z} = \langle c, G \rangle_Z \subset \mathbb{R}^n$ with $G \in \mathbb{R}^{n \times q}$, the F-radius is*

$$\|\mathcal{Z}\|_F := \tfrac{1}{n} \sqrt{ \sum_{i=1}^{n} \sum_{j=1}^{q} G_{(i,j)}^2 }.$$

Please note that in contrast to (Combastel, 2015, Def. 3), we include a normalization factor of $\tfrac{1}{n}$. For set-based training, we compute a gradient set, i.e., the gradient of a function w.r.t. a zonotope, which is also a zonotope, where the center is the derivative w.r.t. the center and the generator matrix is the derivative w.r.t. the generator matrix.

**Definition 3** (Zonotope Gradient)**.** *The gradient of a function $f(\cdot)$ w.r.t. a zonotope $\mathcal{Z} = \langle c, G \rangle_Z \subset \mathbb{R}^n$ is defined as*

$$\nabla_{\mathcal{Z}} f(\mathcal{Z}) := \langle \nabla_c f(\mathcal{Z}), \nabla_G f(\mathcal{Z}) \rangle_Z.$$

### 2.4 Formal Verification of Neural Networks

In this work, we consider the robustness of neural networks for classification tasks: Each dimension of an output $y \in \mathbb{R}^{n_\kappa}$ corresponds to a class, and the dimension with the maximum value determines the predicted class. An input $x \in \mathbb{R}^{n_0}$ is correctly classified by a neural network if the predicted class matches the target label $l \in [n_\kappa]$:

$$\arg\max_{k \in [n_\kappa]} y_{(k)} = l. \tag{5}$$

We call a neural network (locally) robust for a given set of inputs if the neural network correctly classifies every input within the set. Following previous works (Madry et al., 2018; Gowal et al., 2019; Zhang et al., 2019; Müller et al., 2023), we use the $\ell_\infty$-ball of radius $\epsilon \in \mathbb{R}_{>0}$ around an input $x \in \mathbb{R}^{n_0}$ as an input set:

$$\pi_\epsilon(x) \coloneqq \langle x, \epsilon\, I_{n_0} \rangle_Z = \{\, \tilde{x} \in \mathbb{R}^{n_0} \mid \|\tilde{x} - x\|_\infty \leq \epsilon \,\}. \tag{6}$$

For a classification task with target label $l \in [n_\kappa]$, the unsafe set contains every incorrect classification, i.e., there is a dimension $k \in [n_\kappa]$ for which the output $y_{(k)}$ is larger than the output of the target dimension $y_{(l)}$ (Ladner & Althoff, 2023, Prop. B.2):

$$\mathcal{U}_l \coloneqq \{\, y \in \mathbb{R}^{n_\kappa} \mid \exists k \in [n_\kappa]\colon y_{(k)} > y_{(l)} \,\} = \{\, y \in \mathbb{R}^{n_\kappa} \mid (I_{n_\kappa} - e_l^\top \mathbf{1})\, y > 0 \,\}, \tag{7}$$

where $e_l \in \mathbb{R}^{n_\kappa}$ is the $l$-th standard basis vector. For an input set $\mathcal{X} = \pi_\epsilon(x) \subset \mathbb{R}^{n_0}$, we formally verify the robustness of a neural network $\Phi_\theta$ by using set-based computations to efficiently compute (in polynomial time) an enclosure $\mathcal{Y} \subset \mathbb{R}^{n_\kappa}$ of its output set $\mathcal{Y}^* \coloneqq \Phi_\theta(\mathcal{X}) \subseteq \mathcal{Y}$ (Fig. 2): If $\mathcal{Y}$ does not contain unsafe outputs, we have formally verified the neural network for the input set $\mathcal{X}$, as also $\mathcal{Y}^*$ does not intersect with $\mathcal{U}$:

$$\mathcal{Y} \cap \mathcal{U} = \emptyset \implies \mathcal{Y}^* \cap \mathcal{U} = \emptyset. \tag{8}$$

To compute an enclosure $\mathcal{Y}$, we evaluate the layers (Def. 1) over sets. The output set of a linear layer is computed with an affine map (Ladner & Althoff, 2023, Sec. 2.4), whereas the output set of a nonlinear layer is enclosed as it cannot be computed exactly for zonotopes, because they are not closed under nonlinear maps. The required steps are summarized in Fig. 3. The activation function is applied element-wise; hence, the input dimensions are considered independently. We first project the input set onto a dimension and compute bounds (Steps 1 & 2). The activation function is approximated using a linear function, and to ensure the soundness, a bound on the approximation errors is computed (Steps 3 & 4). Finally, the approximation is evaluated over the input set, and the approximation errors are added (Steps 5 & 6).

We define the following set-based forward propagation.

**Proposition 6** (Set-Based Forward Prop., (Ladner & Althoff, 2023, Sec. 2.4))**.** *For an input set $\mathcal{X} \subset \mathbb{R}^{n_0}$, an enclosure $\mathcal{Y} \subset \mathbb{R}^{n_\kappa}$ of the output set $\mathcal{Y}^* \coloneqq \Phi_\theta(\mathcal{X})$ of a neural network can be computed as*

$$\mathcal{H}_0 = \mathcal{X}, \qquad \mathcal{H}_k = \begin{cases} W_k\, \mathcal{H}_{k-1} + b_k & \text{if } k\text{-th layer is linear,} \\ \texttt{enclose}(\phi_k, \mathcal{H}_{k-1}) & \text{otherwise,} \end{cases} \quad \text{for } k \in [\kappa], \qquad \mathcal{Y} = \mathcal{H}_\kappa.$$

*The operation $\texttt{enclose}(L_k, \mathcal{H}_{k-1})$ encloses the image of a nonlinear layer (Ladner & Althoff, 2023, Prop. 2.14). We denote the enclosure of the output set of a neural network $\Phi_\theta$ by $\mathcal{Y} = \texttt{enclose}(\Phi_\theta, \mathcal{X})$.*

Using the enclosure of the output set (Prop. 6), we can formally verify the robustness of a neural network.

**Proposition 7** (Neural Network Verification, (Ladner & Althoff, 2023, Sec. 2.4))**.** *Given a neural network $\Phi\colon \mathbb{R}^{n_0} \to \mathbb{R}^{n_\kappa}$ and an input $x \in \mathbb{R}^{n_0}$ with target label $l \in [n_\kappa]$, we enclose the output set $\mathcal{Y} = \texttt{enclose}(\Phi_\theta, \pi_\epsilon(x))$. We can formally verify the robustness of the neural network by*

$$\max_{y \in \mathcal{Y}} (I_{n_\kappa} - e_l^\top \mathbf{1})\, y \leq 0 \implies \mathcal{Y}^* \cap \mathcal{U} = \emptyset.$$

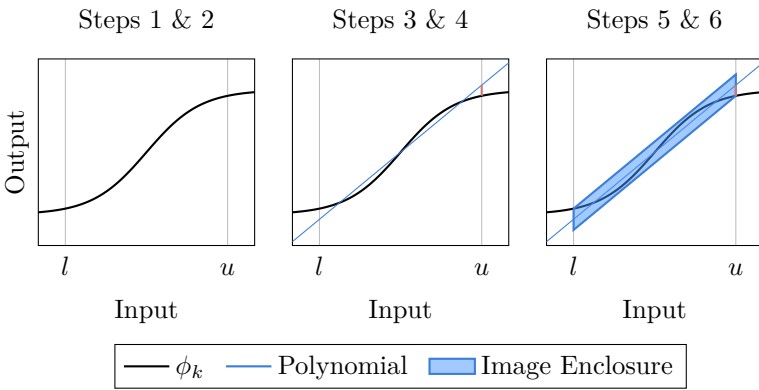

Figure 3: Main steps of an image enclosure (Ladner & Althoff, 2023, Prop. 2.14).

## 2.5 Problem Statement

We want to leverage set-based training to create robust neural networks.

**Definition 4** (Set-Based Training). *Given a set-based loss function $\mathcal{L} : \mathbb{R}^{n_\kappa} \times 2^{\mathbb{R}^{n_\kappa}} \to \mathbb{R}$, set-based training trains a neural network using a gradient set $\nabla_{\mathcal{Y}} \mathcal{L}(t, \mathcal{Y}) \subset \mathbb{R}^{n_\kappa}$ (Def. 3), which defines a different gradient $g \in \nabla_{\mathcal{Y}} \mathcal{L}(t, \mathcal{Y})$ for each point $y \in \mathcal{Y}$.*

Set-based training can directly enforce smaller output sets by choosing gradients that point toward the center of an output set (Fig. 4). Thereby, we can simultaneously improve the robustness of the neural network and simplify the formal verification. We want to use set-based training to train the parameters $\theta$ of a robust neural network $\Phi_\theta$:

$$\min_{\theta} \sum_{(x_i, t_i) \in \mathcal{D}} \mathcal{L}\big(t_i, \texttt{enclose}(\Phi_\theta, \pi_\epsilon(x_i))\big).$$

By reducing the size of the output set using set-based training, we can verify neural networks with fast (polynomial time) algorithms.

# 3 Set-Based Training of Neural Networks

We present a novel set-based algorithm to train robust neural networks with a gradient set (Def. 4). In each training iteration, we (i) enclose the output set of the neural network by zonotopes (Sec. 4.1), (ii) compute a gradient set derived from a set-based loss function using features of the output enclosure (Sec. 3.1), (iii) backpropagate the gradient set (Sec. 4.2), and (iv) aggregate the gradient set to update the parameters of the neural network (Sec. 4.3).

## 3.1 Set-Based Loss Function and Gradient Set

We derive the gradient set used for training by computing the gradient of a set-based loss function, which maps an output set to a loss value (Def. 4). In this work, we define the set-based loss function $\mathcal{L} : \mathbb{R}^{n_\kappa} \times 2^{\mathbb{R}^{n_\kappa}} \to \mathbb{R}$ so that it combines (i) a standard loss of the center of the enclosure (for accuracy) with (ii) the F-radius (Prop. 5) of the enclosure to approximate its size (for robustness) (Fig. 1c).

**Definition 5** (Set-Based Loss). *We define a set-based loss function as*

$$\mathcal{L}(t, \mathcal{Y}) \coloneqq \underbrace{(1 - \tau)\, L(t, c_\kappa)}_{\text{Accuracy Loss}} + \underbrace{\tau/\epsilon\, \|\mathcal{Y}\|_F}_{\text{Robustness Loss}} \quad,$$

*where $L : \mathbb{R}^{n_\kappa} \times \mathbb{R}^{n_\kappa} \to \mathbb{R}$ is a loss function, $\epsilon \in \mathbb{R}_{>0}$ is the perturbation radius, and $\mathcal{Y} = \langle c_\kappa, G_\kappa \rangle_Z$ is an output set.*

Figure 4: Gradients of the F-radius of a zonotope.

The set-based loss function balances the standard loss of the center and the F-radius of the output set using a hyperparameter $\tau \in [0, 1]$, controlling the tradeoff between accuracy and robustness. To make tuning the hyperparameter $\tau$ easier, the F-radius in Def. 5 is normalized using the input perturbation radius $\epsilon \in \mathbb{R}_{>0}$. The normalization is derived by taking the ratio of the F-radii of output enclosure $\mathcal{Y}$ and input set $\mathcal{X} = \pi_\epsilon(x)$, i.e., $\|\mathcal{Y}\|_F / \|\pi_\epsilon(x)\|_F \overset{(6)}{=} 1/\epsilon \, \|\mathcal{Y}\|_F$.

Our set-based loss generalizes the well-established tradeoff-loss (Zhang et al., 2019, Eq. 5), which combines a standard training loss with a boundary loss; for $y = \Phi_\theta(x)$ and weighting factor $\lambda$:

$$L_{\mathrm{TRADES}}(t, y) = \underbrace{L(t, y)}_{\text{standard training loss}} + \max_{\tilde{x} \in \pi_\epsilon(x)} \underbrace{1/\lambda \, L(y, \Phi_\theta(\tilde{x}))}_{\text{boundary loss}}. \qquad (9)$$

The F-radius in our set-based loss and the boundary loss in equation 9 have the same goal of training the robustness of the neural network. However, the F-radius captures the size of an output set in all directions, whereas the boundary loss only considers the distance to the furthest output. Moreover, through the sound set-based computations (Prop. 6), the set-based loss accurately over-approximates the size of the output set. In contrast, the boundary loss is only approximated in (Zhang et al., 2019) using adversarial attacks.

We derive the gradient set for training by computing the gradient of the set-based loss function (Def. 5), which requires the gradient of the F-radius.

**Proposition 8** (Gradient of F-Radius). *Given an output set $\mathcal{Y} = \langle c_\kappa, G_\kappa \rangle_Z \subset \mathbb{R}^{n_\kappa}$, the gradient of the F-radius is*

$$\nabla_{\mathcal{Y}} \|\mathcal{Y}\|_F = \left\langle \mathbf{0}, \frac{1}{n_\kappa \|\mathcal{Y}\|_F} \, G_\kappa \right\rangle_Z.$$

*Proof.* See Appendix D. □

The negative gradients of the F-radius of a zonotope point toward the center of the zonotope (Fig. 4); hence, minimizing the F-radius of a zonotope reduces the size of the zonotope. With Prop. 8, we compute the gradient set used for training:

**Proposition 9** (Set-Based Loss Gradient). *Given an output set $\mathcal{Y} = \langle c_\kappa, G_\kappa \rangle_Z \subset \mathbb{R}^{n_\kappa}$, the gradient of the set-based loss function $\mathcal{L}$ is*

$$\nabla_{\mathcal{Y}} \mathcal{L}(t, \mathcal{Y}) = \left\langle (1 - \tau) \, \nabla_{c_\kappa} L(t, c_\kappa), \frac{\tau}{\epsilon} \, \frac{1}{n_\kappa \|\mathcal{Y}\|_F} \, G_\kappa \right\rangle_Z.$$

*Proof.* See Appendix D. □

Fig. 1c visualizes the gradient set for samples from the output set. For effective set-based training using the gradient set, we require an efficient and differentiable set propagation, which we address next.

## 4 Fast, Batch-wise, and Differentiable Set-Propagation

We require efficient, batch-wise, and differentiable set propagation to implement set-based training. In this section, we first define a forward propagation with the required properties (Sec. 4.1) before we derive the corresponding set-based backpropagation (Sec. 4.2) and weight updates (Sec. 4.3). Finally, we investigate the time complexity of our set-based training (Sec. 4.4).

### 4.1 Set-Based Forward Propagation

The set propagation through linear layers is computed with an affine map, which can be efficiently implemented batch-wise using matrix multiplications on a GPU. Further, we want to efficiently compute batch-wise image enclosures of nonlinear layers. Sampling-based methods (Ladner & Althoff, 2023; Kochdumper et al., 2023) are impractical for this task because they use polynomial regression and thus are not efficient enough for backpropagation. In contrast, (Singh et al., 2018) derives fast analytical solutions for the approximation errors of a specific linear approximation of s-shaped activation functions; however, this approach causes large approximation errors. During training, large approximation errors induce an over-regularization, leading to poor performance (Müller et al., 2023). To address this issue, we derive analytical solutions for the approximation errors of linear approximations for three typical activation functions: rectified linear unit (ReLU), hyperbolic tangent, and logistic sigmoid. Secondly, we provide a linear approximation whose approximation errors are smaller or equal to Singh's enclosure (Singh et al., 2018, Thm. 3.2) while being equally fast to compute.

For the remainder of this section, let $\phi \colon \mathbb{R} \to \mathbb{R}$ be a monotonically increasing function, which is approximated by a linear function $p \colon \mathbb{R} \to \mathbb{R}$, $x \mapsto m\,x + d$ within an interval $[l, u] \subset \mathbb{R}$. The approximation errors of $p$ are given by the largest lower distance and upper distance between $\phi$ and $p$ (Fig. 5).

**Definition 6** (Approximation Error of Linear Approximation). *The approximation error of a linear approximation $p$ for $\phi$ within the interval $[l, u]$ are defined as*

$$\underline{e} := \min_{x \in [l,u]} \phi(x) - p(x), \qquad\qquad \overline{e} := \max_{x \in [l,u]} \phi(x) - p(x).$$

We enclose the output of the activation function $\phi$ as,

$$\phi([l, u]) \subseteq p(x) \oplus [\underline{e}, \overline{e}].$$

Fig. 5 illustrates a linear approximation $p$ of the hyperbolic tangent along with its approximation errors.

**Efficient Computation of Approximation Errors** We efficiently find the approximation errors $\underline{e}$ and $\overline{e}$ of a linear approximation $p$ by only checking specific points of the interval $[l, u]$, i.e., extreme points of the activation function $\phi$.

**Proposition 10** (Approximation Errors for ReLU, Hyperbolic Tangent, and Logistic Sigmoid). *The approximation error of $p$ for $\phi \in \{\mathrm{ReLU}, \tanh, \sigma\}$ are computed as*

$$\underline{e} = \min_{x \in \mathcal{P}_\phi \cap [l,u]} \phi(x) - p(x), \qquad\qquad \overline{e} = \max_{x \in \mathcal{P}_\phi \cap [l,u]} \phi(x) - p(x),$$

*where*

$$\mathcal{P}_{\mathrm{ReLU}} = \{0, l, u\}, \qquad \mathcal{P}_{\tanh} = \left\{\pm \tanh^{-1}\!\left(\sqrt{1-m}\right), l, u\right\}, \qquad \mathcal{P}_{\sigma} = \left\{\pm 2\,\tanh^{-1}\!\left(\sqrt{1-4\,m}\right), l, u\right\}.$$

*Proof.* See Appendix D. □

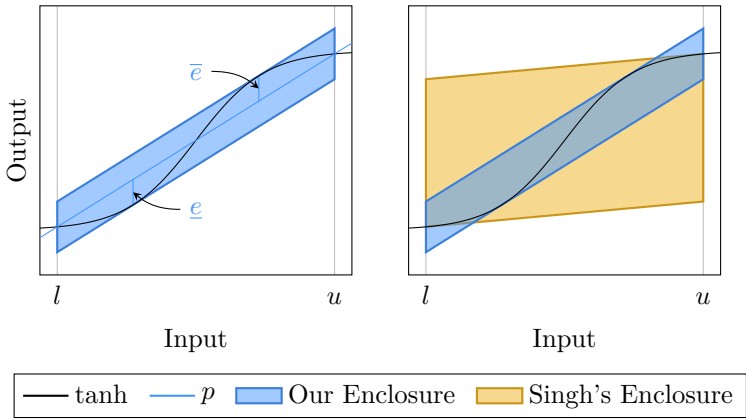

Figure 5: Image enclosure of hyperbolic tangent: (left) Our linear approximation and approximation errors; (right) Comparison of our image enclosure and Singh's enclosure (Singh et al., 2018, Thm. 3.2).

We note that our computation of the approximation errors works for any (monotonically increasing) linear approximation. Moreover, we observe that the offset $d$ of the linear approximation $p$ does not affect the image enclosure; hence, w.l.o.g. we set $d = 0$.

**Definition 7** (Linear Approximation of an Activation Function). *Within the interval $[l, u]$, we approximate an activation function $\phi$ by a linear function $p(x) := m\,x$, where*

$$m := \frac{\phi(u) - \phi(l)}{u - l}.$$

We note that the slope of our approximation is identical to the slope used for the triangle relaxation by DeepPoly (Singh et al., 2019a); however, instead of formulating linear constraints, we require the explicit computation (Prop. 10) of the approximation errors for the zonotope propagation. Our image enclosure no longer uses a polynomial regression or requires sampling to compute the approximation errors (Kochdumper et al., 2023, Sec. 3.2). Moreover, only matrix operations are required, i.e., matrix multiplication, matrix addition, min, and max. Therefore, it can be efficiently computed batch-wise, taking full advantage of GPU acceleration.

**Our Enclosure vs. Singh's Enclosure**  We motivate the choice for our image enclose by comparing it with Singh's enclosure (Singh et al., 2018, Thm. 3.2). For s-shaped activation functions, e.g., hyperbolic tangent and logistic sigmoid, we prove that the approximation error of our linear approximation (Def. 7) is always smaller or equal to the approximation error of Singh's enclosure (Singh et al., 2018, Thm. 3.2) w.r.t. the area in the input-output plane (see Fig. 5) measuring the integrated approximation error over $[l, u]$:

$$\text{area}([\underline{e}, \overline{e}], [l, u]) := (u - l)\,(\overline{e} - \underline{e}). \tag{10}$$

**Proposition 11.** *Let $\phi$ be an s-shaped function, and let $[l, u]$ be an interval. Moreover, let $[\underline{e}, \overline{e}]$ be the approximation errors of our enclosure (Def. 6 and 7), and let $e_S$ be the approximation error of Singh's enclosure (Singh et al., 2018, Thm. 3.2). It holds that*

$$\text{area}([\underline{e}, \overline{e}], [l, u]) \leq \text{area}([-e_S, e_S], [l, u]).$$

*Proof.* See Appendix D. □

Fig. 6 shows an instance where the output set computed with our image enclosure is significantly smaller and does not intersect the unsafe region. In contrast, the output set computed with Singh's enclosure intersects the unsafe region.

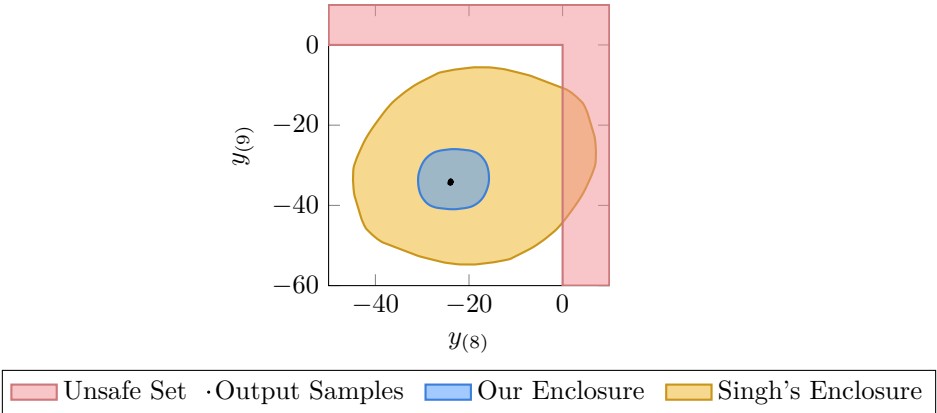

Figure 6: Comparison of the output set of a neural network computed with our image enclosures and Singh's enclosure (Singh et al., 2018, Thm. 3.2).

## 4.2 Set-Based Backpropagation

We now derive the corresponding set-based backpropagation. Analogous to the standard backpropagation (Prop. 1), the set-based backpropagation computes for every layer of the neural network the gradient set w.r.t. the output set $\mathcal{H}_k = \langle c_k, G_k \rangle_Z$:

$$\mathcal{G}_k = \langle c'_k, G'_k \rangle_Z \coloneqq \nabla_{\mathcal{H}_k} \mathcal{L}(t, \mathcal{Y}). \tag{11}$$

The set-based backpropagation of linear layers is straightforward as it applies an affine map (Prop. 1). However, the backpropagation of nonlinear layers is more involved: while the image enclosure only uses linear approximations, these depend on the input set. Thus, intuitively, we have to apply the product rule.

Please note that, we use the plus symbol $(+)$ between zonotopes to denote their element-wise addition: $\langle c_1, G_1 \rangle_Z + \langle c_2, G_2 \rangle_Z = \langle c_1 + c_2, G_1 + G_2 \rangle_Z$, whereas the Minkowski sum is denoted as $\langle c_1, G_1 \rangle_Z \oplus \langle c_2, G_2 \rangle_Z = \langle c_1 + c_2, [G_1 \; G_2] \rangle_Z$.

**Proposition 12** (Backpropagation through Image Enclosure). *Assume the $k$-th layer is a nonlinear layer with activation function $\phi_k$. Given an input set $\mathcal{H}_{k-1} = \langle c_{k-1}, G_{k-1} \rangle_Z$ with $G_{k-1} \in \mathbb{R}^{n_{k-1} \times p}$ and a gradient set $\mathcal{G}_k = \langle c'_k, G'_k \rangle_Z$, the gradient set $\mathcal{G}_{k-1} = \langle c'_{k-1}, G'_{k-1} \rangle_Z$ is computed for each dimension $i \in [n_k]$ as*

$$\mathcal{G}_{k-1(i)} = m_{k(i)} \, \mathcal{G}_{k(i)} + \left( c'_{k(i)} \, c_{k-1(i)} + G'_{k(i,[p])} \, G^{\top}_{k-1(i,\cdot)} \right) \nabla_{\mathcal{H}_{k-1(i)}} m_{k(i)}$$
$$+ \frac{1}{2} \left( c'_{k(i)} + G'_{k(i,p+i)} \right) \nabla_{\mathcal{H}_{k-1(i)}} \overline{e}_{k(i)} + \frac{1}{2} \left( c'_{k(i)} - G'_{k(i,p+i)} \right) \nabla_{\mathcal{H}_{k-1(i)}} \underline{e}_{k(i)}.$$

*The operation $\mathcal{G}_{k-1} = \mathtt{backpropEnclose}(\phi_k, \mathcal{G}_k)$ computes the gradient set of an image enclosure.*

*Proof.* See Appendix D. $\qquad\square$

Using the backpropagation of an image enclosure, we can (analogous to Prop. 1) backpropagate the gradient sets through all layers of a neural network.

**Proposition 13** (Set-Based Backpropagation). *The gradient sets $\mathcal{G}_k$ are computed in reverse order as*

$$\mathcal{G}_\kappa = \nabla_{\mathcal{Y}} \mathcal{L}(t, \mathcal{Y}), \qquad \mathcal{G}_{k-1} = \begin{cases} W_k^{\top} \, \mathcal{G}_k & \textit{if } k\textit{-th layer is linear,} \\ \mathtt{backpropEnclose}(\phi_k, \mathcal{G}_k) & \textit{otherwise,} \end{cases}$$

*for $k = \kappa, \kappa - 1, \ldots, 1$.*

*Proof.* See Appendix D. $\qquad\square$

### 4.3 Set-Based Update of Weights and Biases

We now describe how to use the gradient set $\mathcal{G}_k$ and the set of inputs $\mathcal{H}_{k-1}$ to update the weights and biases of a linear layer: Intuitively, we compute the outer product between the gradient set $\mathcal{G}_k$ and input set $\mathcal{H}_{k-1}$. To avoid clutter, we define the outer product between two zonotopes $\mathcal{Z}_1 = \langle c_1, G_1 \rangle_Z \subset \mathbb{R}^{n_1}$ and $\mathcal{Z}_2 = \langle c_2, G_2 \rangle_Z \subset \mathbb{R}^{n_2}$ as $\mathcal{Z}_1 \odot \mathcal{Z}_2^\top \coloneqq c_1\, c_2^\top + G_1\, G_2^\top \in \mathbb{R}^{n_1 \times n_2}$.

**Proposition 14** (Gradient Set of Weights and Biases)**.** *The gradients of the set-based loss w.r.t. a weight matrix and a bias vector are*

$$\nabla_{W_k}\mathcal{L}(t, \mathcal{Y}) = \nabla_{\mathcal{H}_k}\mathcal{L}(t, \mathcal{Y}) \odot \mathcal{H}_{k-1}^\top, \qquad\qquad \nabla_{b_k}\mathcal{L}(t, \mathcal{Y}) = \nabla_{\mathcal{H}_k}\mathcal{L}(t, \mathcal{Y}) \odot \langle 1, \mathbf{0} \rangle_Z^\top .$$

*Proof.* See Appendix D. □

The weight matrices and bias vectors are updated analogous to standard training (4) using the gradients of the set-based loss function:

$$W_k \leftarrow W_k - \eta\, \nabla_{W_k}\mathcal{L}(t, \mathcal{Y}), \qquad\qquad b_k \leftarrow b_k - \eta\, \nabla_{b_k}\mathcal{L}(t, \mathcal{Y}). \qquad (12)$$

### 4.4 Computational Complexity

Finally, we derive the time complexity of set-based training. Alg. 1 implements our image enclosure.

---

**Algorithm 1:** Image Enclosure of a Nonlinear Layer.

**1** **function** enclose$(\phi_k, \mathcal{H}_{k-1})$
**2** $\quad$ Find bounds $[l_{k-1}, u_{k-1}]$ of $\mathcal{H}_{k-1}$ $\hfill$ // Prop. 2
**3** $\quad$ **for** $i \leftarrow 1$ **to** $n_k$ **do**
**4** $\quad\quad$ Find linear approx. $m_{k(i)}\, x$ of $\phi_k$ $\hfill$ // Def. 7
**5** $\quad\quad$ Find approx. errors $\underline{e}_{k(i)}, \overline{e}_{k(i)}$ $\hfill$ // Prop. 10
**6** $\quad$ $\widetilde{\mathcal{H}}_k \leftarrow \mathrm{Diag}(m_k)\, \mathcal{H}_{k-1} + \tfrac{1}{2}\left(\overline{e}_k + \underline{e}_k\right)$ $\hfill$ // Prop. 4
**7** $\quad$ $\mathcal{H}_k \leftarrow \widetilde{\mathcal{H}}_k \oplus [\underline{e}_k, \overline{e}_k]$ $\hfill$ // Prop. 3
**8** $\quad$ **return** $\mathcal{H}_k$

---

We provide the time complexity of Alg. 1 w.r.t. the number of input dimensions and generators.

**Proposition 15** (Time Complexity of Alg. 1)**.** *For an input set $\mathcal{H}_{k-1} = \mathcal{Z}$ with $c \in \mathbb{R}^n$ and $G \in \mathbb{R}^{n \times q}$, Alg. 1 has time complexity $\mathcal{O}(n^2\, q)$ w.r.t. the number of input dimensions $n$ and the number of generators $q$.*

*Proof.* See Appendix D. □

Using Prop. 15, we can compute the time complexity of a set-based forward propagation.

**Proposition 16** (Time Complexity of Prop. 6)**.** *The zonotopes during a forward propagation (Prop. 6) have at most $q \le n_0 + \sum_{k \in [\kappa]} n_k$ generators. Let $n_{max} \coloneqq \max_{k \in [\kappa]} n_k$ be the maximum number of neurons in a layer. The set-based forward propagtion (Prop. 6) has time complexity $\mathcal{O}(n_{max}^2\, q\, \kappa)$ w.r.t. $n_{max}$, $q$ and the number of layers $\kappa$.*

*Proof.* See Appendix D. □

Alg. 2 implements an iteration of set-based training. First, we enclose the output set of the neural network (lines 1–6). Then, we compute the gradient set (line 8), which is backpropagated through the neural network (lines 9–15). Finally, the parameters of the neural network are updated (lines 12–13).

**Proposition 17** (Time Complexity of Alg. 2)**.** *Alg. 2 has time complexity $\mathcal{O}(n_{max}^2\, q\, \kappa)$ w.r.t. $n_{max}$, $q$ and the number of layers $\kappa$.*

---

**Algorithm 2:** Set-based Training Iteration. Hyperparameters: $\epsilon \in \mathbb{R}_{>0}$, $\tau \in [0,1]$, and $\eta \in \mathbb{R}_{>0}$.

**Data:** Input $x \in \mathbb{R}^{n_0}$, Target $t \in \mathbb{R}^{n_\kappa}$
**Result:** Neural Network with Updated Parameters

**1** $\mathcal{H}_0 \leftarrow \langle x, \epsilon I_{n_0} \rangle_Z$       // construct input set (6)
**2** **for** $k \leftarrow 1$ **to** $\kappa$ **do**       // set-based forward prop. (Prop. 6)
**3**     **if** *k-th layer is linear* **then**
**4**       $\mathcal{H}_k \leftarrow W_k\,\mathcal{H}_{k-1} + b_k$
**5**     **else**
**6**       $\mathcal{H}_k \leftarrow \texttt{enclose}(\phi_k, \mathcal{H}_{k-1})$

**7** $\mathcal{Y} \leftarrow \mathcal{H}_\kappa$       // obtain output set (6)
**8** $\mathcal{G}_\kappa \leftarrow \nabla_{\mathcal{Y}}\mathcal{L}(t, \mathcal{Y})$       // compute gradient set (Prop. 9)
**9** **for** $k \leftarrow \kappa$ **to** $1$ **do**       // set-based backprop. (Prop. 13)
**10**     **if** *k-th layer is linear* **then**
**11**       $\mathcal{G}_{k-1} \leftarrow W_k^\top\,\mathcal{G}_k$
**12**       $W_k \leftarrow W_k - \eta\,(\mathcal{G}_k \odot \mathcal{H}_{k-1}^\top)$       // Prop. 14 and equation 12
**13**       $b_k \leftarrow b_k - \eta\,(\mathcal{G}_k \odot \langle 1, \mathbf{0} \rangle_Z^\top)$       // Prop. 14 and equation 12
**14**     **else**
**15**       $\mathcal{G}_{k-1} \leftarrow \texttt{backpropEnclose}(\phi_k, \mathcal{G}_k)$       // Prop. 12

---

*Proof.* See Appendix D.       □

The time complexity of set-based training is polynomial, and compared to point-based training, only has an additional factor $q \in \mathcal{O}(n_0 + \sum_{k \in [\kappa]} n_k)$. The increased time complexity is expected because set-based training propagates entire generator matrices through the neural network. Moreover, for some linear relaxation methods, similar time complexities are reported (Zhang et al., 2018).

## 5 Evaluation

We use the MATLAB toolbox CORA (Althoff, 2015) to implement set-based training. Following previous works (Gowal et al., 2019), we train a 6-layer convolutional neural network on MNIST (LeCun et al., 2010), SVHN (Netzer et al., 2011), CIFAR-10 (Krizhevsky, 2009), and TINYIMAGENET (Le & Yang, 2015). The training details can be found in Appendix A. We first present the main results and then justify our design choices with extensive ablation studies.

### 5.1 Main Results

We compare our set-based training against adversarial tradeoff training, i.e., TRADES (Zhang et al., 2019), as well as two state-of-the-art interval-based training approaches, i.e., IBP (Gowal et al., 2019) and SABR (Müller et al., 2023). For each training scheme, we report the clean accuracy (percentage of correctly classified test samples), the falsified accuracy (percentage of test samples for which PGD finds an adversarial example) (Gowal et al., 2019), and, unlike previous works, the fast-verified accuracy (percentage of verified test samples using zonotopes). Most related works report verified accuracies computed with slow (exponential time) verification algorithms using branch-and-bound (Balunović & Vechev, 2020; Müller et al., 2023; Mao et al., 2023) or mixed-integer programming (Gowal et al., 2019); the verification can take up to 34h for MNIST networks (Müller et al., 2023, App. C). Our goal is the fast (polynomial time) verification of neural networks. Thus, we report fast-verified accuracy, which uses a single zonotope propagation (Prop. 6) with polynomial time complexity (Prop. 16) for the verification of each test input without a branch-and-bound procedure. The verification of a 6-layer convolutional neural network trained on MNIST with $10\,000$ test inputs takes approximately 30min.

Table 1: Comparison with state of the art (mean & std. dev. of the best 3 runs across 5 seeds).

| Dataset | $\epsilon_\infty$ | Method | clean Acc. | falsified Acc. | fast-verified Acc. (max) | |
|---|---|---|---|---|---|---|
| MNIST | 0.1 | TRADES | **99.40** $\pm$ 0.04 | **98.40** $\pm$ 0.06 | 0.00 $\pm$ 0.00 | (0.00) |
| | | IBP | 97.76 $\pm$ 0.26 | 96.32 $\pm$ 0.39 | 95.58 $\pm$ 0.41 | (96.05) |
| | | SABR | 97.94 $\pm$ 0.44 | 95.84 $\pm$ 1.28 | 92.80 $\pm$ 3.57 | (95.86) |
| | | Set (ours) | 98.76 $\pm$ 0.29 | 97.52 $\pm$ 0.26 | **95.89** $\pm$ 0.82 | (**96.40**) |
| CIFAR-10 | $2/255$ | TRADES | **82.96** $\pm$ 0.35 | **67.35** $\pm$ 0.01 | 0.00 $\pm$ 0.00 | (0.00) |
| | | IBP | 46.54 $\pm$ 2.92 | 42.13 $\pm$ 2.56 | 36.83 $\pm$ 1.74 | (38.80) |
| | | SABR | 54.20 $\pm$ 0.90 | 48.04 $\pm$ 0.49 | **39.96** $\pm$ 0.30 | (**40.30**) |
| | | Set (ours) | 63.87 $\pm$ 2.23 | 55.17 $\pm$ 1.56 | 37.79 $\pm$ 1.55 | (39.36) |
| SVHN | 0.01 | TRADES | **91.30** $\pm$ 0.28 | **78.34** $\pm$ 0.28 | 0.00 $\pm$ 0.00 | (0.00) |
| | | IBP | 72.33 $\pm$ 4.26 | 61.27 $\pm$ 3.46 | **46.93** $\pm$ 3.09 | (**50.18**) |
| | | SABR | 81.79 $\pm$ 4.36 | 63.26 $\pm$ 14.32 | 12.62 $\pm$ 6.63 | (17.81) |
| | | Set (ours) | 82.66 $\pm$ 3.13 | 72.35 $\pm$ 2.68 | 39.60 $\pm$ 2.39 | (41.35) |
| TINYIMAGENET | $1/255$ | TRADES | **28.14** $\pm$ 0.56 | **19.80** $\pm$ 0.43 | 0.00 $\pm$ 0.00 | (0.00) |
| | | IBP | 9.62 $\pm$ 1.17 | 8.76 $\pm$ 0.95 | 2.89 $\pm$ 1.27 | (4.07) |
| | | SABR | 13.74 $\pm$ 1.38 | 12.49 $\pm$ 1.25 | **3.68** $\pm$ 0.87 | (**4.68**) |
| | | Set (ours) | 16.70 $\pm$ 2.79 | 14.80 $\pm$ 2.54 | 0.02 $\pm$ 0.02 | (0.05) |

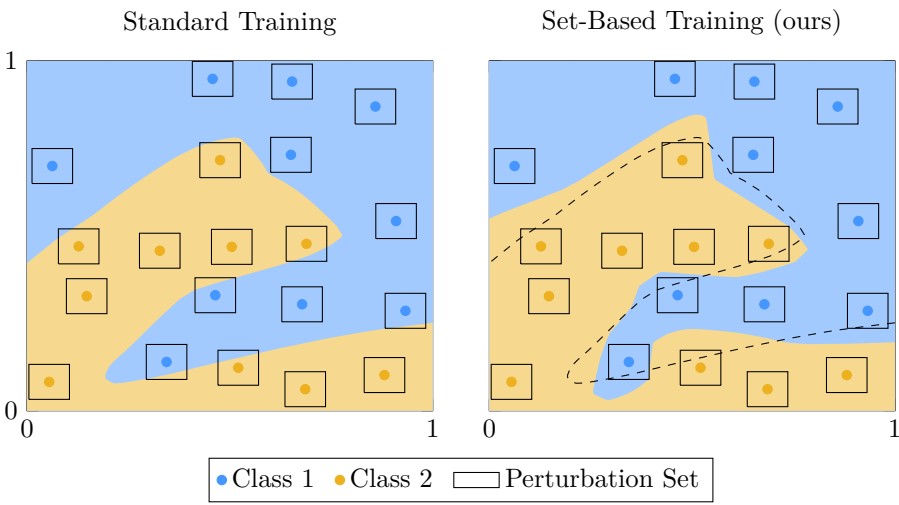

Figure 7: Comparing the decision bounds of point-based (left) and set-based (right) training. The dashed line is the decision boundary of point-based training.

Tab. 1 shows our results. Across all datasets, TRADES shows the best performance for clean and falsified accuracy; however, the verification is notoriously hard. Both interval-based approaches, IBP and SABR, have lower clean and falsified accuracy than TRADES but are significantly easier to verify. SABR achieves slightly better performance due to less regularization by propagating smaller intervals. For SVHN, SABR has lower verified accuracy than IBP, indicating that stronger verification algorithms are required. Our set-based training strikes a balance between TRADES and interval-based approaches, with significantly higher verified accuracy compared to TRADES and higher clean and falsified accuracy compared to IBP and SABR.

Table 2: Performance on MNIST for different weighting factors in the set-based loss (Def. 5).

| Method | clean Acc. | falsified Acc. | fast-verified Acc. (max) | |
|---|---|---|---|---|
| Set ($\tau = 0.0$) | $\mathbf{98.87} \pm 0.01$ | $96.48 \pm 0.18$ | $89.40 \pm 0.58$ | $(90.04)$ |
| Set ($\tau = 0.01$) | $98.74 \pm 0.07$ | $\mathbf{96.77} \pm 0.29$ | $93.45 \pm 0.47$ | $(93.73)$ |
| Set ($\tau = 0.1$) | $97.57 \pm 0.09$ | $95.60 \pm 0.17$ | $\mathbf{94.03} \pm 0.04$ | $(\mathbf{94.06})$ |
| Set ($\tau = 0.2$) | $97.17 \pm 0.05$ | $95.07 \pm 0.12$ | $93.70 \pm 0.12$ | $(93.82)$ |
| Set ($\tau = 0.3$) | $96.59 \pm 0.07$ | $93.46 \pm 1.20$ | $88.91 \pm 4.36$ | $(93.95)$ |

Table 3: Performance on MNIST of different input sets.

| Method | clean Acc. | fast-verified Acc. (max) | |
|---|---|---|---|
| Set ($\ell_\infty$) | $\mathbf{97.58} \pm 0.03$ | $93.83 \pm 0.06$ | $(93.89)$ |
| Set (FGSM) | $97.57 \pm 0.09$ | $\mathbf{94.03} \pm 0.04$ | $(\mathbf{94.06})$ |

## 5.2 Understanding Robustness

To better understand the robustness of neural networks, we compare the learned decision boundaries of standard (point-based) and set-based training for a simple binary classification task (Fig. 7). Both training methods learn the training data perfectly, but set-based training pushes the decision boundaries away from the samples, making the set-based-trained model more robust. For some samples, the decision boundary of the point-based trained neural network crosses their perturbation sets, which is not the case for the set-based-trained neural network.

## 5.3 Ablation Studies

We conduct ablation studies to justify our design choices. Please refer to Appendix A for details.

**Weighting of Robustness Loss** It has been shown that there is a fundamental tradeoff between accuracy and robustness (Zhang et al., 2019). The weighting factor $\tau$ in our set-based loss (Def. 5) can trade off accuracy with robustness. Tab. 2 shows the accuracies for different values of $\tau$: For smaller values of $\tau$, the clean accuracy is larger, and the verified accuracy is larger for larger values. Thus, this hyperparameter can be used to tune the tradeoff between accuracy and robustness.

**Input Set** Compared to other robust training approaches (Gowal et al., 2019; Müller et al., 2023; Mao et al., 2023), our set-based training is not limited to multi-dimensional intervals as input sets. In Tab. 3, we compare using the $\ell_\infty$-ball as an input set with a smaller input zonotope where the generators are computed using adversarial attacks (see Appendix A for details). The smaller input set leads to better performance; thereby, we confirm the observations from previous works (Müller et al., 2023) that smaller input sets reduce the regularization by creating smaller approximation errors and thus resulting in better performance.

**Set-Propagation Method** We compare the propagation of zonotopes with our image enclosure to other set propagation methods during our set-based training (Tab. 4): IBP, zonotopes with Singh's enclosure, and zonotopes with our enclosure (Prop. 6). The results support our choice of enclosure as it consistently obtains higher accuracies than the other enclosures. This is due to the smaller accumulated approximation errors (Prop. 11). Thus, we observe that our enclosure produces the best results, justifying the use of zonotopes for our image enclosure.

**Size of the Output Sets and Robustness** In Tab. 5, we compare the size of the output sets (of the first 1000 test samples) and the Lipschitz constant of the best performing neural networks trained on MNIST. The Lipschitz constant of a neural network is another metric for robustness, because it bounds the sensitivity of the neural network for input changes (Fazlyab et al., 2019). Moreover, we compare the interval norm (Althoff,

Table 4: Performance on MNIST of different set propagation methods.

| Method | clean Acc. | falsified Acc. | fast-verified Acc. (max) | |
|---|---|---|---|---|
| Set (Zonotope+ours) | $\mathbf{98.60} \pm 0.21$ | $\mathbf{97.13} \pm 0.13$ | $\mathbf{96.13} \pm 0.13$ | ($\mathbf{96.25}$) |
| Set (Zonotope+Singh) | $98.36 \pm 0.13$ | $96.60 \pm 0.59$ | $94.44 \pm 2.57$ | ($96.04$) |
| Set (IBP) | $97.77 \pm 0.19$ | $96.11 \pm 0.20$ | $95.35 \pm 0.27$ | ($95.62$) |

Table 5: Comparing the sizes of the output sets and Lipschitz for different training methods.

| Method | Size (Interval Norm) | Lipschitz Constant |
|---|---|---|
| TRADES | $276.36 \pm 120.40$ | $\mathbf{9.72 \times 10^{7}}$ |
| IBP | $4.77 \pm 1.96$ | $2.78 \times 10^{16}$ |
| SABR | $5.26 \pm 2.39$ | $4.20 \times 10^{17}$ |
| Set (ours) | $\mathbf{1.85} \pm 1.27$ | $3.49 \times 10^{12}$ |

2023, Sec. 3.1) of the output set, because computing the volume of a zonotope is computationally hard. The Lipschiz constant of the TRADES-trained network is the smallest, which explains the great empirical robustness (falsified Acc. in Tab. 1); however, the output sets are too large for verification (fast-verified Acc. in Tab. 1). Our set-based-trained neural network produces the smallest output sets and a smaller Lipschitz constant compared to IBP and SABR.

## 6 Conclusion

This paper introduces the first set-based training procedure for neural networks that uses gradient sets: During training, we enclose the output set of the neural network using set propagation and derive a gradient set, which contains a different gradient for each possible output. By choosing gradients that point to the center of the output set, we can directly reduce the size of the output set. Thereby, we can simultaneously improve robustness and simplify the formal verification. The set-based training is made possible by a fast, batch-wise, and differentiable propagation of zonotopes, which utilizes analytical solutions for approximation errors. Our experimental results demonstrate that our set-based approach effectively trains robust neural networks, which have competitive performance and admit fast verification (in polynomial time). Thereby, we demonstrate that gradient sets can be effectively used to train robust neural networks. Hence, set-based training represents a promising new direction for robust neural network training and a significant step toward fast verification of neural networks and, thus, the widespread adoption of formal verification of neural networks.

## Acknowledgements

This work was partially supported by the project SPP-2422 (No. 500936349) and the project FAI (No. 286525601), both funded by the German Research Foundation (DFG). We also want to thank our colleague Mark Wetzlinger from our research group for his revisions of the manuscript.

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

## A    Evaluation Details

**Hardware**  Our experiments were run on a server with $2\times$AMD EPYC 7763 (64 cores/128 threads), 2TB RAM, and a NVIDIA A100 40GB GPU.

**Training Hyperparameters**  The training hyperparameters are listed in Tab. 6 and the neural network architectures in Tab. 7. We use the same notation as (Gowal et al., 2019): Conv $k$ $w \times h + s$ denotes a convolutional layer with $k$ filters of size $w \times h$ and stide $s$, and Fc $n$ denotes a fully connected layer with $n$ neurons. The weights and biases are initialized as in (Shi et al., 2021). We use Adam optimizer (Kingma & Ba, 2015) with the recommended hyperparameters. For all training methods, we tried to use hyperparameters as close as possible to the reported hyperparameters in their respective paper: for (Gowal et al., 2019) we use $\kappa = 1/2$, for (Zhang et al., 2019) we use $1/\lambda = 6$, for (Müller et al., 2023) we use $\lambda = 0.1$ for Cifar-10, and $\lambda = 0.4$ for Mnist and Svhn. For any PGD during training (used by methods (Madry et al., 2018; Zhang et al., 2019; Müller et al., 2023)) we used the settings from (Müller et al., 2023): 8 iterations with an initial step size 0.5, which is decayed twice by 0.1 at iterations 4 and 7. All PGD attacks for testing are computed with 40 iterations of step size 0.01. Moreover, all reported accuracies are averaged over the best 3 of 5 runs. We clip gradients with a $\ell_2$-norm of greater than 10. All reported perturbation radii are w.r.t. normalized inputs between 0 and 1. To reduce computational resources, we performed the ablation studies with a smaller 3-layer convolutional neural network trained on Mnist.

Table 6: Training hyperparameters.

| Dataset | $\eta$ | $\epsilon$ | $\tau$ | Batch Size | #Epochs (*warm-up / ramp-up*) | Decay |
|---------|--------|-----------|--------|------------|-------------------------------|-------|
| MNIST | $5 \cdot 10^{-4}$ | $0.1$ | $0.1$ | $256$ | $70\,(1/20)$ | $50, 60$ |
| CIFAR-10 | $5 \cdot 10^{-4}$ | $2/255$ | $0.005$ | $128$ | $160\,(40/120)$ | $120, 140$ |
| SVHN | $5 \cdot 10^{-4}$ | $0.01$ | $0.01$ | $128$ | $70\,(20/50)$ | $50, 60$ |
| TINYIMAGENET | $5 \cdot 10^{-4}$ | $1/255$ | $0.1$ | $64$ | $70\,(20/50)$ | $50, 60$ |

Table 7: Neural network architectures. Each linear layer or convolutional layer (except the last layer) is followed by a batch normalization and a nonlinear activation layer (ReLU).

| CNN3 | CNN6 |
|------|------|
| CONV $5\ 4 \times 4 + 2$ | CONV $32\ 3 \times 3 + 1$ |
| CONV $10\ 4 \times 4 + 1$ | CONV $32\ 4 \times 4 + 2$ |
| FC $100$ | CONV $64\ 3 \times 3 + 1$ |
| | CONV $64\ 4 \times 4 + 2$ |
| | FC $512$ |
| | FC $512$ |

**Datasets** MNIST contains $60\,000$ grayscale images of size $28 \times 28$. Each image depicts a handwritten digit from 0 to 9. SVHN is a real-world dataset that contains $73\,257$ colored images of digits of house numbers that are cropped to size $32 \times 32$. The CIFAR-10 dataset contains $60\,000$ colored images of size $32 \times 32$. The TINYIMAGENET data set contains $100\,000$ colored images of size $64 \times 64$ labeled with 200 classes. We use the canonical split of training and test data for each dataset and the entire test data for evaluation; because test labels are not available for TINYIMAGENET, we follow (Müller et al., 2023) and use the validation set for testing. Following (Xu et al., 2020), we augment the CIFAR-10 and TINYIMAGENET dataset with random crop and flips. The perturbation is applied before the normalization to ensure comparability with the literature.

**Improvements for Scalability** The complexity of our set-based training depends on the number of generators of the output set used during training (Prop. 17). We use two methods to reduce the number of generators: (i) We propagate the approximation errors as intervals through the neural network and compute the Minkowski sum at the end. (ii) We use input sets with fewer number of generators. Our set-based training can use any zonotopic input set and is not limited to $\ell_\infty$-input sets. Building on previous work (Müller et al., 2023), we use adversarial attacks to construct smaller input sets that focus on critical regions of the input; thereby, the regularization through large approximation errors is reduced. Previous works only use a single adversarial attack to construct a smaller $\ell_\infty$-input set (Müller et al., 2023). We extend this idea and use several adversarial attacks to construct zonotopic input sets: We shift the center of the input set to the average attack and use scaled directions of the attacks as generators for the input set. Given an input $x \in \mathbb{R}^{n_0}$, we compute $\lambda$ adversarial attacks $\tilde{x}_i = x + \delta_i$ for $i \in [\lambda]$ (e.g., using FGSM). The input set is constructed as

$$\mathcal{X} = \left\langle x + 1/\lambda \sum_{\lambda}^{i=1} \delta_i,\ 1/\lambda \begin{bmatrix} \delta_1 & \delta_2 & \cdots & \delta_\lambda \end{bmatrix} \right\rangle_Z.$$

**Fairness** We note that compared to the literature, we use smaller neural networks and compare the means across several training runs; most literature only reports their best-observed scores, which is problematic regarding reproducibility and fairness because it is unclear how many training seeds were used. Therefore, to create a fair comparison, we have reimplemented related approaches (Madry et al., 2018; Gowal et al., 2019; Zhang et al., 2019; Müller et al., 2023). The implementations have been validated by reproducing their reported results.

Table 8: Comparing the maximum perturbation radius.

| Method | $\epsilon$ **(max)** | |
|---|---|---|
| TRADES | $0.0715 \pm 0.0140$ | $(0.1094)$ |
| IBP | $\mathbf{0.1856} \pm 0.0528$ | $(0.2441)$ |
| SABR | $0.1792 \pm 0.0482$ | $(0.2422)$ |
| Set (ours) | $0.1782 \pm 0.0494$ | $(\mathbf{0.2676})$ |

Table 9: Performance on CIFAR10 for different weighting factors in the set-based loss (Def. 5).

| Method | clean Acc. | falsified Acc. | fast-verified Acc. (max) | |
|---|---|---|---|---|
| Set ($\tau = 0.0$) | $\mathbf{61.42} \pm 1.12$ | $50.80 \pm 0.62$ | $31.98 \pm 0.61$ | $(32.53)$ |
| Set ($\tau = 0.005$) | $59.95 \pm 1.13$ | $\mathbf{50.89} \pm 0.64$ | $38.50 \pm 0.32$ | $(38.87)$ |
| Set ($\tau = 0.01$) | $59.29 \pm 0.99$ | $50.37 \pm 0.76$ | $39.59 \pm 0.31$ | $(39.92)$ |
| Set ($\tau = 0.05$) | $56.75 \pm 0.79$ | $48.80 \pm 0.46$ | $41.10 \pm 0.13$ | $(41.24)$ |
| Set ($\tau = 0.1$) | $54.90 \pm 0.84$ | $47.90 \pm 0.58$ | $\mathbf{41.57} \pm 0.28$ | $(\mathbf{41.87})$ |
| Set ($\tau = 0.2$) | $50.53 \pm 1.18$ | $45.05 \pm 0.83$ | $40.08 \pm 0.58$ | $(40.49)$ |
| Set ($\tau = 0.3$) | $45.55 \pm 1.77$ | $41.29 \pm 1.51$ | $36.13 \pm 2.97$ | $(38.23)$ |

## B  Additional Experiments

**Maximum Perturbation Radius**  In Tab. 8, we compare the maximal perturbation radius for which we can verify the robustness. We used binary search (10 iterations) to compute the maximum perturbation radius for the first 1000 test samples of the MNIST dataset. The perturbation radii of IBP, SABR, and Set are comparable and significantly higher compared to TRADES.

**Weighting of Robustness Loss**  Tab. 9 shows the accuracies for different values of $\tau$ of CNN3 trained on the CIFAR10 dataset.

**Set-Propagation Method**  Tab. 10 shows the results of CNN6 trained on MNIST for different set propagation methods.

**Training Times**  The training times are compared in Tab. 11. The computational overhead of set-based training is slightly higher than TRADES, IBP, and SABR. However, the TRADES-trained neural networks have great empirical performance (clean and falsified accuracy Tab. 1) but cannot be verified using polynomial time verification approaches (fast-verified accuracy Tab. 1). IBP and SABR have significantly lower empirical performance compared to TRADES. Our set-based trained neural networks have higher empirical performance compared to IBP and SABR and have great verified robustness (fast-verified accuracy Tab. 1). Moreover, with our set-based training, we can use a weighting parameter in the set-based loss to explicitly tune the tradeoff between empirical performance and verified robustness. Therefore, set-based training strikes a balance between training time, empirical performance, and verified robustness.

## C  Reproducibility of Fig. 7

Fig. 7 compares the decision boundaries of a point-based and a set-based training of a neural network for a binary classification task. The network architecture is nn-med: 5 layers with 100 neurons each. The training

Table 10: Performance of CNN6 on MNIST of different set propagation methods.

| Method | clean Acc. | falsified Acc. | fast-verified Acc. (max) | |
|---|---|---|---|---|
| Set (Zonotope+ours) | **98.66** ± 0.02 | **97.71** ± 0.08 | **97.08** ± 0.08 | (**97.15**) |
| Set (Zonotope+Singh) | 98.13 ± 0.88 | 97.61 ± 0.19 | 96.88 ± 0.19 | (97.08) |
| Set (IBP) | 98.08 ± 0.08 | 96.60 ± 0.21 | 95.96 ± 0.21 | (96.11) |

Table 11: Training time with CNN6 on MNIST (min. of 5 runs) [sec / Epoch].

| Method | Training Time [sec / Epoch]($\downarrow$) |
|---|---|
| Point | 6.1 |
| TRADES | 49.6 |
| IBP | 19.9 |
| SABR | 52.0 |
| Set (ours) | 61.2 |

data are 20 random input samples $x_i \in [0,1]^2$ with corresponding targets $t_i \in \{0,1\}^2$:

$$\mathcal{D} = \left\{ \left( \begin{bmatrix} 0.0622 \\ 0.6995 \end{bmatrix}, \begin{bmatrix} 0 \\ 1 \end{bmatrix} \right), \left( \begin{bmatrix} 0.6534 \\ 0.9409 \end{bmatrix}, \begin{bmatrix} 0 \\ 1 \end{bmatrix} \right), \left( \begin{bmatrix} 0.4759 \\ 0.7163 \end{bmatrix}, \begin{bmatrix} 1 \\ 0 \end{bmatrix} \right), \left( \begin{bmatrix} 0.8812 \\ 0.1020 \end{bmatrix}, \begin{bmatrix} 1 \\ 0 \end{bmatrix} \right), \left( \begin{bmatrix} 0.5047 \\ 0.4685 \end{bmatrix}, \begin{bmatrix} 1 \\ 0 \end{bmatrix} \right), \right.$$
$$\left( \begin{bmatrix} 0.1470 \\ 0.3275 \end{bmatrix}, \begin{bmatrix} 1 \\ 0 \end{bmatrix} \right), \left( \begin{bmatrix} 0.3439 \\ 0.1395 \end{bmatrix}, \begin{bmatrix} 0 \\ 1 \end{bmatrix} \right), \left( \begin{bmatrix} 0.9098 \\ 0.5422 \end{bmatrix}, \begin{bmatrix} 0 \\ 1 \end{bmatrix} \right), \left( \begin{bmatrix} 0.8588 \\ 0.8696 \end{bmatrix}, \begin{bmatrix} 0 \\ 1 \end{bmatrix} \right), \left( \begin{bmatrix} 0.0545 \\ 0.0825 \end{bmatrix}, \begin{bmatrix} 1 \\ 0 \end{bmatrix} \right),$$
$$\left( \begin{bmatrix} 0.6889 \\ 0.4771 \end{bmatrix}, \begin{bmatrix} 1 \\ 0 \end{bmatrix} \right), \left( \begin{bmatrix} 0.9329 \\ 0.2857 \end{bmatrix}, \begin{bmatrix} 0 \\ 1 \end{bmatrix} \right), \left( \begin{bmatrix} 0.6781 \\ 0.3043 \end{bmatrix}, \begin{bmatrix} 0 \\ 1 \end{bmatrix} \right), \left( \begin{bmatrix} 0.4641 \\ 0.3302 \end{bmatrix}, \begin{bmatrix} 0 \\ 1 \end{bmatrix} \right), \left( \begin{bmatrix} 0.4575 \\ 0.9487 \end{bmatrix}, \begin{bmatrix} 0 \\ 1 \end{bmatrix} \right),$$
$$\left( \begin{bmatrix} 0.1272 \\ 0.4699 \end{bmatrix}, \begin{bmatrix} 1 \\ 0 \end{bmatrix} \right), \left( \begin{bmatrix} 0.6506 \\ 0.7315 \end{bmatrix}, \begin{bmatrix} 0 \\ 1 \end{bmatrix} \right), \left( \begin{bmatrix} 0.5207 \\ 0.1229 \end{bmatrix}, \begin{bmatrix} 1 \\ 0 \end{bmatrix} \right), \left( \begin{bmatrix} 0.3271 \\ 0.4574 \end{bmatrix}, \begin{bmatrix} 1 \\ 0 \end{bmatrix} \right), \left. \left( \begin{bmatrix} 0.6858 \\ 0.0616 \end{bmatrix}, \begin{bmatrix} 1 \\ 0 \end{bmatrix} \right) \right\}.$$

We train both neural networks for 200 epochs with a mini-batch size of 10 using the Adam optimizer with a learning rate of $\eta = 0.01$. For set-based training, we use $\epsilon = 0.05$ with $\tau = 0.1$.

## D Proofs

**Proposition 8.** *Given an output set* $\mathcal{Y} = \langle c_\kappa, G_\kappa \rangle_Z \subset \mathbb{R}^{n_\kappa}$, *the gradient of the F-radius is*

$$\nabla_{\mathcal{Y}} \|\mathcal{Y}\|_F = \left\langle \mathbf{0}, \frac{1}{n_\kappa \|\mathcal{Y}\|_F} G_\kappa \right\rangle_Z.$$

*Proof.* The center does not affect the F-radius, hence $\nabla_{c_\kappa} \|\mathcal{Y}\|_F = \mathbf{0}$. The F-radius is the sum of all squared entries of the generator matrix. Hence,

$$\nabla_{G_\kappa} \|\mathcal{Y}\|_F = \frac{1}{n_\kappa} \nabla_{G_\kappa} \sqrt{\mathbf{1}^\top (G_\kappa \odot G_\kappa) \mathbf{1}} = \frac{G_\kappa}{n_\kappa \|\mathcal{Y}\|_F}. \tag{13}$$

Thus,

$$\nabla_{\mathcal{Y}} \|\mathcal{Y}\|_F \stackrel{\text{Def. 3}}{=} \langle \nabla_{c_\kappa} \|\mathcal{Y}\|_F, \nabla_{G_\kappa} \|\mathcal{Y}\|_F \rangle_Z \stackrel{(13)}{=} \left\langle \mathbf{0}, \frac{G_\kappa}{n_\kappa \|\mathcal{Y}\|_F} \right\rangle_Z = \frac{1}{n_\kappa \|\mathcal{Y}\|_F} \langle \mathbf{0}, G_\kappa \rangle_Z. \qquad \square$$

**Proposition 9** (Set-Based Loss Gradient). *Given an output set* $\mathcal{Y} = \langle c_\kappa, G_\kappa \rangle_Z \subset \mathbb{R}^{n_\kappa}$, *the gradient of the set-based loss function* $\mathcal{L}$ *is*

$$\nabla_{\mathcal{Y}} \mathcal{L}(t, \mathcal{Y}) = \left\langle (1 - \tau) \nabla_{c_\kappa} L(t, c_\kappa), \frac{\tau}{\epsilon} \frac{1}{n_\kappa \|\mathcal{Y}\|_F} G_\kappa \right\rangle_Z.$$

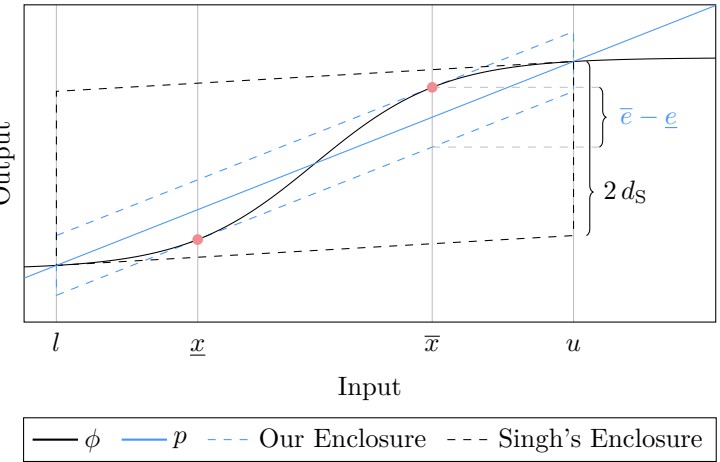

Figure 8: Illustration for Prop. 11.

*Proof.* This follows from Def. 5 and Prop. 8:

$$\nabla_{\mathcal{Y}}\mathcal{L}(t,\mathcal{Y}) \overset{\text{Def. 5}}{=} (1-\tau)\,\nabla_{\mathcal{Y}}L(t,c_\kappa) + \frac{\tau}{\epsilon}\,\nabla_{\mathcal{Y}}\|\mathcal{Y}\|_F \overset{\text{Def. 3}}{=} (1-\tau)\,\langle\nabla_{c_\kappa}L(t,c_\kappa),\mathbf{0}\rangle_Z + \frac{\tau}{\epsilon}\,\nabla_{\mathcal{Y}}\|\mathcal{Y}\|_F$$

$$\overset{\text{Prop. 8}}{=} (1-\tau)\,\langle\nabla_{c_\kappa}L(t,c_\kappa),\mathbf{0}\rangle_Z + \frac{\tau}{\epsilon\,n_\kappa\,\|\mathcal{Y}\|_F}\,\langle\mathbf{0},G_\kappa\rangle_Z$$

$$= \left\langle (1-\tau)\,\nabla_{c_\kappa}L(t,c_\kappa),\,\frac{\tau}{\epsilon\,n_\kappa\,\|\mathcal{Y}\|_F}\,G_\kappa \right\rangle_Z. \qquad \square$$

**Proposition 10.** *The approximation error of $p$ for $\phi \in \{\mathrm{ReLU}, \tanh, \sigma\}$ are computed as*

$$\underline{e} = \min_{x\in\mathcal{P}_\phi\cap[l,u]} \phi(x) - p(x), \qquad\qquad \overline{e} = \max_{x\in\mathcal{P}_\phi\cap[l,u]} \phi(x) - p(x),$$

*where*

$$\mathcal{P}_{\mathrm{ReLU}} = \{0, l, u\}, \qquad \mathcal{P}_{\tanh} = \{\pm\tanh^{-1}(\sqrt{1-m}), l, u\}, \qquad \mathcal{P}_\sigma = \{\pm 2\tanh^{-1}(\sqrt{1-4m}), l, u\}.$$

*Proof. Case (i).* $\phi = \mathrm{ReLU}$. $\mathrm{ReLU}(x) - p(x)$ is linear for $[l,0]$ and $[0,u]$. Thus, the approximation errors are found at the bounds $x \in \{l, u\}$ or where $0 \in [l, u]$ at $x = 0$.

*Case (ii).* $\phi = \tanh$. The derivative of the hyperbolic tangent is $\tanh'(x) = 1 - \tanh(x)^2$. To compute the extreme points of $\tanh(x) - p(x)$, we demand that its derivative is 0 and simplify the terms:

$$0 = \mathrm{d}/\mathrm{d}x(\tanh(x) - p(x)) \qquad\Leftrightarrow\qquad 0 = 1 - \tanh(x)^2 - m \qquad\Leftrightarrow\qquad x = \pm\tanh^{-1}(\sqrt{1-m}).$$

*Case (iii).* $\phi = \sigma$. To compute the extreme points of $\sigma(x) - p(x)$, we demand that its derivative is 0 and simplify the terms:

$$0 = \mathrm{d}/\mathrm{d}x(\sigma(x) - p(x)) \quad\Leftrightarrow\quad 0 = \mathrm{d}/\mathrm{d}x(1/2\,(\tanh(x/2) + 1) - p(x)) \quad\Leftrightarrow\quad x = \pm 2\tanh^{-1}(\sqrt{1-4m}). \;\square$$

**Proposition 11.** *Let $\phi$ be an s-shaped function, and let $[l,u]$ be an interval. Moreover, let $[\underline{e},\overline{e}]$ be the approximation errors of our enclosure (Def. 6 and 7), and let $e_S$ be the approximation error of Singh's enclosure (Singh et al., 2018, Thm. 3.2). It holds that*

$$\mathrm{area}([\underline{e},\overline{e}],[l,u]) \leq \mathrm{area}([-e_S, e_S],[l,u]).$$

*Proof.* We first observe that the approximation errors $\underline{e}$ and $\overline{e}$ of $p$ can be computed at points $\overline{x}, \underline{x} \in [l, u]$ such that $\underline{x} \leq \overline{x}$ (Fig. 8):

$$\underline{e} = \phi(\underline{x}) - p(\underline{x}), \qquad\qquad\qquad \overline{e} = \phi(\overline{x}) - p(\overline{x}). \qquad (14)$$

Singh's enclosure (Singh et al., 2018, Thm. 3.2) uses the linear approximation $p_S(x) = m_S\,x + d_S$, with slope $m_S$, offset $d_S$, and approximation error $e_S$:

$$m_S = \min(\phi'(l), \phi'(u)), \quad d_S = \tfrac{1}{2}\,(\phi(u) + \phi(l) - m_S\,(u + l)), \quad e_S = \tfrac{1}{2}\,(\phi(u) - \phi(l) - m_S\,(u - l)). \quad (15)$$

Using Def. 7, we obtain the following inequality:

$$m = \frac{\phi(u) - \phi(l)}{u - l} \geq m_S.$$

Hence, we have

$$m\,(\overline{x} - \underline{x}) \geq m_S\,(\overline{x} - \underline{x}). \qquad (16)$$

Moreover, from (15) we have for all $x \in [l, u]$:

$$\begin{aligned}
\phi(x) - \phi(l) \geq m_S\,(x - l) &\implies \phi(\underline{x}) \geq \phi(l) + m_S\,(\underline{x} - l), \\
\phi(u) - \phi(x) \geq m_S\,(u - x) &\implies \phi(\overline{x}) \leq \phi(u) - m_S\,(u - \overline{x}).
\end{aligned} \qquad (17)$$

Ultimately, we have

$$\begin{aligned}
\overline{e} - \underline{e} &\stackrel{(14)}{=} \phi(\overline{x}) - p(\overline{x}) - (\phi(\underline{x}) - p(\underline{x})) \stackrel{\text{Def. 7}}{=} \phi(\overline{x}) - (m\,\overline{x}) - (\phi(\underline{x}) - (m\,\underline{x})) \\
&\stackrel{(16)}{\leq} \phi(\overline{x}) - \phi(\underline{x}) - m_S\,(\overline{x} - \underline{x}) \stackrel{(17)}{\leq} \phi(u) - m_S\,(u - \overline{x}) - (\phi(l) + m_S\,(\underline{x} - l)) - m_S\,(\overline{x} - \underline{x}) \stackrel{(15)}{=} 2\,e_S.
\end{aligned} \qquad (18)$$

Hence, we obtain the bound $\overline{e} - \underline{e} \leq 2\,e_S$. Thus,

$$\text{area}([\underline{e}, \overline{e}], [l, u]) \stackrel{(10)}{=} (u - l)\,(\overline{e} - \underline{e}) \stackrel{(18)}{\leq} (u - l)\,2\,e_S \stackrel{(10)}{=} \text{area}([-e_S, e_S], [l, u]).$$

$\square$

**Proposition 12** (Backpropagation through Image Enclosure). *Assume the k-th layer is a nonlinear layer with activation function $\phi_k$. Given an input set $\mathcal{H}_{k-1} = \langle c_{k-1}, G_{k-1}\rangle_Z$ with $G_{k-1} \in \mathbb{R}^{n_{k-1} \times p}$ and a gradient set $\mathcal{G}_k = \langle c'_k, G'_k\rangle_Z$, the gradient set $\mathcal{G}_{k-1} = \langle c'_{k-1}, G'_{k-1}\rangle_Z$ is computed for each dimension $i \in [n_k]$ as*

$$\begin{aligned}
\mathcal{G}_{k-1(i)} = {}& m_{k(i)}\,\mathcal{G}_{k(i)} + \left(c'_{k(i)}\,c_{k-1(i)} + G'_{k(i,[p])}\,G^\top_{k-1(i,\cdot)}\right) \nabla_{\mathcal{H}_{k-1(i)}} m_{k(i)} \\
& + \frac{1}{2}\left(c'_{k(i)} + G'_{k(i,p+i)}\right) \nabla_{\mathcal{H}_{k-1(i)}} \overline{e}_{k(i)} + \frac{1}{2}\left(c'_{k(i)} - G'_{k(i,p+i)}\right) \nabla_{\mathcal{H}_{k-1(i)}} \underline{e}_{k(i)}.
\end{aligned}$$

*The operation $\mathcal{G}_{k-1} = \texttt{backpropEnclose}(\phi_k, \mathcal{G}_k)$ computes the gradient set of an image enclosure.*

*Proof.* The image enclosure adds $n_k$ generators, hence the input set $\mathcal{H}_{k-1}$ has $n_k$ generators less than gradient set $\mathcal{G}_k$, i.e. $G_{k-1} \in \mathbb{R}^{n_k \times p}$ and $G'_k \in \mathbb{R}^{n_k \times q}$ with $q = p + n_k$.

We unfold (11) and rewrite the gradient set $\mathcal{G}_{k-1}$ using the chain rule for partial derivatives. Let $\mathcal{H}_k = \langle c_k, G_k\rangle_Z$ with $G_k \in \mathbb{R}^{n_k \times q}$ be the output set of the $k$-th layer and let $\mathcal{G}_k = \langle c'_k, G'_k\rangle_Z$ be the gradient set w.r.t. $\mathcal{H}_k$:

$$\begin{aligned}
\mathcal{G}_{k-1} &\stackrel{(11)}{=} \nabla_{\mathcal{H}_{k-1}} \mathcal{L}(t, \mathcal{Y}) = \sum_{i=1}^{n_k} \frac{\partial \mathcal{L}(t, \mathcal{Y})}{\partial c_{k(i)}}\, \nabla_{\mathcal{H}_{k-1}} c_{k(i)} + \sum_{i=1}^{n_k} \sum_{j=1}^{q} \frac{\partial \mathcal{L}(t, \mathcal{Y})}{\partial G_{k(i,j)}}\, \nabla_{\mathcal{H}_{k-1}} G_{k(i,j)} \\
&= \sum_{i=1}^{n_k} c'_{k(i)}\, \nabla_{\mathcal{H}_{k-1}} c_{k(i)} + \sum_{i=1}^{n_k} \sum_{j=1}^{q} G'_{k(i,j)}\, \nabla_{\mathcal{H}_{k-1}} G_{k(i,j)}.
\end{aligned} \qquad (19)$$

We split (19) into three summands:

$$\mathcal{G}_{k-1,c} = \sum_{i=1}^{n_k} c'_{k(i)} \; \nabla_{\mathcal{H}_{k-1}} c_{k(i)},$$

$$\mathcal{G}_{k-1,p} = \sum_{i=1}^{n_k} \sum_{j=1}^{p} G'_{k(i,j)} \; \nabla_{\mathcal{H}_{k-1}} G_{k(i,j)},$$

$$\mathcal{G}_{k-1,q} = \sum_{i=1}^{n_k} \sum_{j=p+1}^{q} G'_{k(i,j)} \; \nabla_{\mathcal{H}_{k-1}} G_{k(i,j)}.$$

Hence,

$$\mathcal{G}_{k-1} = \mathcal{G}_{k-1,c} + \mathcal{G}_{k-1,p} + \mathcal{G}_{k-1,q}. \tag{20}$$

Furthermore, the input set $\mathcal{H}_{k-1}$ is enclosed by the interval $[l_{k-1}, u_{k-1}]$, where $l_{k-1} = c_{k-1} - |G_{k-1}| \, \mathbf{1}$ and $u_{k-1} = c_{k-1} + |G_{k-1}| \, \mathbf{1}$ (Prop. 2).

Firstly, we derive the gradient $\nabla_{\mathcal{H}_{k-1}} c_{k(i)}$ needed for $\mathcal{G}_{k-1,c}$, for which we need the gradients of center $c_{k(i)} = m_{k(i)} \, c_{k-1(i)} + 1/2 \left( \overline{e}_{k(i)} + \underline{e}_{k(i)} \right)$ w.r.t. the input set $\mathcal{H}_{k-1}$ for each dimension $i \in [n_k]$. The image enclosure is applied for each dimension individually; therefore, we can consider each dimension separately because for any dimensions $i, j \in [n_{k-1}]$, where $i \neq j$:

$$\nabla_{\mathcal{H}_{k-1(j)}} c_{k(i)} = \langle 0, \mathbf{0} \rangle_Z, \qquad\qquad \nabla_{\mathcal{H}_{k-1(j)}} G_{k(i,\cdot)} = \langle 0, \mathbf{0} \rangle_Z. \tag{21}$$

Let $i \in [n_k]$ be a fixed dimension and $j \in [p]$ a fixed index of a generator. We require the gradient of the slope $m_{k(i)}$ and the offset $e_{c,k(i)}$. The gradient of the slope $m_{k(i)}$ is:

$$\frac{\partial m_{k(i)}}{\partial c_{k-1(i)}} \overset{\text{Def. 7}}{=} \frac{\phi'(u_{k-1(i)}) - \phi'(l_{k-1(i)})}{u_{k-1(i)} - l_{k-1(i)}},$$

$$\frac{\partial m_{k(i)}}{\partial G_{k-1(i,j)}} \overset{\text{Def. 7}}{=} \left( \frac{\phi'(u_{k-1(i)}) + \phi'(l_{k-1(i)}) - 2\, m_{k(i)}}{u_{k-1(i)} - l_{k-1(i)}} \right) \text{sign}(G_{k-1(i,j)}).$$

Let $\overline{x}_k$ and $\underline{x}_k$ be the points of the approximation errors $\overline{e}_k$ and $\underline{e}_k$:

$$\overline{x}_k = \arg\max_{x \in \mathcal{P}} \phi_k(x) - p_k(x), \qquad\qquad \underline{x}_k = \arg\min_{x \in \mathcal{P}} \phi_k(x) - p_k(x).$$

To prevent repetitions in this proof, let $g$ denote the center, or an arbitrary generator of the input set $\mathcal{H}_{k-1}$:

$$g \in \left\{ c_{k-1}, G_{k-1(\cdot,1)}, G_{k-1(\cdot,2)}, \dots, G_{k-1(\cdot,p)} \right\}.$$

For $(x, e) \in \{(\overline{x}_k, \overline{e}_k), (\underline{x}_k, \underline{e}_k)\}$, we apply the chain rule and the product-rule to derive the gradients of the approximation error $e_{(i)}$:

$$\frac{\partial e_{(i)}}{\partial g_{(i)}} = \frac{\partial \left( \phi_k(x_{(i)}) - m_{k(i)} \, x_{(i)} \right)}{\partial g_{(i)}} = \left( \phi'_k(x_{(i)}) - m_{k(i)} \right) \frac{\partial x_{(i)}}{\partial g_{(i)}} - \frac{\partial m_{k(i)}}{\partial g_{(i)}} \, x_{(i)}.$$

Using the gradient of the slope $m_{k(i)}$, we derive the gradient of the center $c_{k(i)}$:

$$\frac{\partial c_{k(i)}}{\partial c_{k-1(i)}} = \frac{\partial \left( m_{k(i)} \, c_{k-1(i)} + \frac{1}{2} \left( \overline{e}_{k(i)} + \underline{e}_{k(i)} \right) \right)}{\partial c_{k-1(i)}} = m_{k(i)} + \frac{\partial m_{k(i)}}{\partial c_{k-1(i)}} \, c_{k-1(i)} + \frac{1}{2} \left( \frac{\partial \overline{e}_{k(i)}}{\partial c_{k-1(i)}} + \frac{\partial \underline{e}_{k(i)}}{\partial c_{k-1(i)}} \right),$$

$$\frac{\partial c_{k(i)}}{\partial G_{k-1(i,j)}} = \frac{\partial \left( m_{k(i)} \, c_{k-1(i)} + \frac{1}{2} \left( \overline{e}_{k(i)} + \underline{e}_{k(i)} \right) \right)}{\partial G_{k-1(i,j)}} = \frac{\partial m_{k(i)}}{\partial G_{k-1(i,j)}} \, c_{k-1(i)} + \frac{1}{2} \left( \frac{\partial \overline{e}_{k(i)}}{\partial G_{k-1(i,j)}} + \frac{\partial \underline{e}_{k(i)}}{\partial G_{k-1(i,j)}} \right).$$

Hence, we have

$$
\begin{aligned}
\nabla_{\mathcal{H}_{k-1(i)}} c_{k(i)} &= \left\langle \frac{\partial c_{k(i)}}{\partial c_{k-1(i)}}, \frac{\partial c_{k(i)}}{\partial G_{k-1(i,\cdot)}} \right\rangle_Z \\
&= m_{k(i)} + c_{k-1(i)} \; \nabla_{\mathcal{H}_{k-1(i)}} m_{k(i)} + \frac{1}{2} \left( \nabla_{\mathcal{H}_{k-1(i)}} \overline{e}_{k(i)} + \nabla_{\mathcal{H}_{k-1(i)}} \underline{e}_{k(i)} \right).
\end{aligned}
\tag{22}
$$

Secondly, we derive the gradient $\nabla_{\mathcal{H}_{k-1(i)}} G_{k(i,j)}$ needed for $\mathcal{G}_{k-1,p}$. Let $j' \in [n_k]$ be a different generator index: $j' \neq j$; the gradient of $G_{k(i,j)}$ is

$$
\begin{aligned}
\frac{\partial G_{k(i,j)}}{\partial c_{k-1(i)}} &= \frac{\partial \big( m_{k(i)} \, G_{k-1(i,j)} \big)}{\partial c_{k-1(i)}} = \frac{\partial m_{k(i)}}{\partial c_{k-1(i)}} \, G_{k-1(i,j)}, \\
\frac{\partial G_{k(i,j)}}{\partial G_{k-1(i,j)}} &= \frac{\partial \big( m_{k(i)} \, G_{k-1(i,j)} \big)}{\partial G_{k-1(i,j)}} = m_{k(i)} + \frac{\partial m_{k(i)}}{\partial G_{k-1(i,j)}} \, G_{k-1(i,j)}, \\
\frac{\partial G_{k(i,j)}}{\partial G_{k-1(i,j')}} &= \frac{\partial \big( m_{k(i)} \, G_{k-1(i,j)} \big)}{\partial G_{k-1(i,j')}} = \frac{\partial m_{k(i)}}{\partial G_{k-1(i,j')}} \, G_{k-1(i,j)}.
\end{aligned}
$$

Hence, we have

$$
\nabla_{\mathcal{H}_{k-1(i)}} G_{k(i,j)} = \left\langle \frac{\partial G_{k(i,j)}}{\partial c_{k-1(i)}}, \frac{\partial G_{k(i,j)}}{\partial G_{k-1(i,\cdot)}} \right\rangle_Z = G_{k-1(i,j)} \; \nabla_{\mathcal{H}_{k-1(i)}} m_{k(i)} + m_{k(i)} \left\langle \mathbf{0}, \mathbb{1}_{(i,j)} \right\rangle_Z,
\tag{23}
$$

where $\mathbb{1}_{(i,j)} \in \{0,1\}$ is a matrix containing only zeros except for position $(i,j)$ which contains a one.

Thirdly, we derive $\mathcal{G}_{k-1,q}$. Please recall, the diagonal entries of $G_{k(\cdot,p+[n_k])}$ contain the approximation errors, while the non-diagonal entries are 0; hence, the gradient of any non-diagonal entry $j' \in [q] : j' > p \wedge j' \neq p+i$ is 0: $\nabla_{\mathcal{H}_{k-1(i)}} G_{k(i,j')} = \left\langle 0, \mathbf{0} \right\rangle_Z$. Hence, we can simplify $\mathcal{G}_{k-1,q}$:

$$
\mathcal{G}_{k-1,q(i)} = \sum_{j=p+1}^{q} G'_{k(i,j)} \; \nabla_{\mathcal{H}_{k-1(i)}} G_{k(i,j)} = G'_{k(i,p+i)} \frac{1}{2} \left( \nabla_{\mathcal{H}_{k-1(i)}} \overline{e}_{k(i)} - \nabla_{\mathcal{H}_{k-1(i)}} \underline{e}_{k(i)} \right).
\tag{24}
$$

We add $\mathcal{G}_{k-1,c}$ and $\mathcal{G}_{k-1,p}$ followed by reordering the terms:

$$
\begin{aligned}
\mathcal{G}_{k-1,c(i)} + \mathcal{G}_{k-1,p(i)} &= c'_{k(i)} \; \nabla_{\mathcal{H}_{k-1(i)}} c_{k(i)} + \sum_{j=1}^{p} G'_{k(i,j)} \; \nabla_{\mathcal{H}_{k-1(i)}} G_{k(i,j)} \\
&\overset{(22)}{=} c'_{k(i)} \left( m_{k(i)} + c_{k-1(i)} \; \nabla_{\mathcal{H}_{k-1(i)}} m_{k(i)} + \frac{1}{2} \left( \nabla_{\mathcal{H}_{k-1(i)}} \overline{e}_{k(i)} + \nabla_{\mathcal{H}_{k-1(i)}} \underline{e}_{k(i)} \right) \right) \\
&\quad + \sum_{j=1}^{p} G'_{k(i,j)} \; \nabla_{\mathcal{H}_{k-1(i)}} G_{k(i,j)} \\
&\overset{(23)}{=} c'_{k(i)} \left( m_{k(i)} + c_{k-1(i)} \; \nabla_{\mathcal{H}_{k-1(i)}} m_{k(i)} + \frac{1}{2} \left( \nabla_{\mathcal{H}_{k-1(i)}} \overline{e}_{k(i)} + \nabla_{\mathcal{H}_{k-1(i)}} \underline{e}_{k(i)} \right) \right) \\
&\quad + \sum_{j=1}^{p} G'_{k(i,j)} \left( G_{k-1(i,j)} \; \nabla_{\mathcal{H}_{k-1(i)}} m_{k(i)} + m_{k(i)} \left\langle \mathbf{0}, \mathbb{1}_{(i,j)} \right\rangle_Z \right) \\
&= m_{k(i)} \, \mathcal{G}_{k(i)} + \left( c_{k-1(i)} \, c'_{k(i)} + G^{\top}_{k-1(i,\cdot)} \, G'_{k(i,[p])} \right) \nabla_{\mathcal{H}_{k-1(i)}} m_{k(i)} \\
&\quad + c'_{k(i)} \frac{1}{2} \left( \nabla_{\mathcal{H}_{k-1(i)}} \overline{e}_{k(i)} + \nabla_{\mathcal{H}_{k-1(i)}} \underline{e}_{k(i)} \right).
\end{aligned}
\tag{25}
$$

Finally, we obtain

$$
\begin{aligned}
\mathcal{G}_{k-1(i)} &\overset{(20)}{=} \mathcal{G}_{k-1,c(i)} + \mathcal{G}_{k-1,p(i)} + \mathcal{G}_{k-1,q(i)} \\
&\overset{(25)}{=} m_{k(i)}\,\mathcal{G}_{k(i)} + \left(c_{k-1(i)}\,c'_{k(i)} + G'_{k(i,[p])}\,G^{\top}_{k-1(i,\cdot)}\right)\nabla_{\mathcal{H}_{k-1(i)}}m_{k(i)} + c'_{k(i)}\,\nabla_{\mathcal{H}_{k-1(i)}}e_{c,k(i)} + \mathcal{G}_{k-1,q(i)} \\
&\overset{(24)}{=} m_{k(i)}\,\mathcal{G}_{k(i)} + \left(c_{k-1(i)}\,c'_{k(i)} + G'_{k(i,[p])}\,G^{\top}_{k-1(i,\cdot)}\right)\nabla_{\mathcal{H}_{k-1(i)}}m_{k(i)} \\
&\quad + c'_{k(i)}\frac{1}{2}\left(\nabla_{\mathcal{H}_{k-1(i)}}\overline{e}_{k(i)} + \nabla_{\mathcal{H}_{k-1(i)}}\underline{e}_{k(i)}\right) + G'_{k(i,p+i)}\frac{1}{2}\left(\nabla_{\mathcal{H}_{k-1(i)}}\overline{e}_{k(i)} - \nabla_{\mathcal{H}_{k-1(i)}}\underline{e}_{k(i)}\right) \\
&= m_{k(i)}\,\mathcal{G}_{k(i)} + \left(c_{k-1(i)}\,c'_{k(i)} + G'_{k(i,[p])}\,G^{\top}_{k-1(i,\cdot)}\right)\nabla_{\mathcal{H}_{k-1(i)}}m_{k(i)} \\
&\quad + \frac{1}{2}\left(c'_{k(i)} + G'_{k(i,p+i)}\right)\nabla_{\mathcal{H}_{k-1(i)}}\overline{e}_{k(i)} + \frac{1}{2}\left(c'_{k(i)} - G'_{k(i,p+i)}\right)\nabla_{\mathcal{H}_{k-1(i)}}\underline{e}_{k(i)}. \qquad \square
\end{aligned}
$$

**Proposition 13** (Set-Based Backpropagation). *The gradient sets $\mathcal{G}_k$ are computed in reverse order as*

$$
\mathcal{G}_{\kappa} = \nabla_{\mathcal{Y}}\mathcal{L}(t,\mathcal{Y}), \qquad\qquad \mathcal{G}_{k-1} = \begin{cases} W_k^{\top}\,\mathcal{G}_k & \text{if } k\text{-th layer is linear,} \\ \texttt{backpropEnclose}(\phi_k,\mathcal{G}_k) & \text{otherwise,} \end{cases}
$$

*for $k = \kappa, \kappa-1, \ldots, 1$.*

*Proof.* If $k = \kappa$, we compute the gradient of the set-based loss according to Prop. 9. We assume $k < \kappa$. Let $\mathcal{G}_k = \langle c'_k, G'_k\rangle_Z$ and $\mathcal{H}_k = \langle c_k, G_k\rangle_Z$.

We split cases on the type of the $k$-th layer and simplify the terms.

*Case (i).* The $k$-th layer is linear. For dimension $i \in [n_k]$, we have

$$
\begin{aligned}
\nabla_{\mathcal{H}_{k-1}}c_{k(i)} &= \nabla_{\mathcal{H}_{k-1}}\left(W_{k(i,\cdot)}\,c_{k-1} + b_{k(i)}\right) = \left\langle W^{\top}_{k(i,\cdot)},\mathbf{0}\right\rangle_Z, \\
\nabla_{\mathcal{H}_{k-1}}G_{k(i,j)} &= \nabla_{\mathcal{H}_{k-1}}\left(W_{k(i,\cdot)}\,G_{k-1(\cdot,j)}\right) = \left\langle \mathbf{0}, W^{\top}_{k(i,\cdot)}\,e_j^{\top}\right\rangle_Z.
\end{aligned} \tag{26}
$$

Thus,

$$
\begin{aligned}
\mathcal{G}_{k-1} &\overset{(19)}{=} \sum_{i=1}^{n_k} c'_{k(i)}\,\nabla_{\mathcal{H}_{k-1}}c_{k(i)} + \sum_{i=1}^{n_k}\sum_{j=1}^{q} G'_{k(i,j)}\,\nabla_{\mathcal{H}_{k-1}}G_{k(i,j)} \\
&\overset{(26)}{=} \sum_{i=1}^{n_k} c'_{k(i)}\left\langle W^{\top}_{k(i,\cdot)},\mathbf{0}\right\rangle_Z + \sum_{i=1}^{n_k}\sum_{j=1}^{q} G'_{k(i,j)}\left\langle \mathbf{0}, W^{\top}_{k(i,\cdot)}\,e_j^{\top}\right\rangle_Z \\
&= \left\langle \sum_{i=1}^{n_k} W^{\top}_{k(i,\cdot)}\,c'_{k(i)}, \sum_{i=1}^{n_k} W^{\top}_{k(i,\cdot)}\,G'_{k(i,\cdot)}\right\rangle_Z \\
&= \left\langle W_k^{\top}\,c'_k, W_k^{\top}\,G'_k\right\rangle_Z \\
&= W_k^{\top}\,\mathcal{G}_k.
\end{aligned}
$$

*Case (ii).* The $k$-th layer is nonlinear. Prop. 12 proves the correctness.

$\square$

**Proposition 14** (Gradient Set w.r.t. Weights and Bias). *The gradients of the set-based loss w.r.t. a weight matrix and a bias vector are*

$$
\nabla_{W_k}\mathcal{L}(t,\mathcal{Y}) = \nabla_{\mathcal{H}_k}\mathcal{L}(t,\mathcal{Y}) \odot \mathcal{H}_{k-1}^{\top}, \qquad\qquad \nabla_{b_k}\mathcal{L}(t,\mathcal{Y}) = \nabla_{\mathcal{H}_k}\mathcal{L}(t,\mathcal{Y}) \odot \langle 1,\mathbf{0}\rangle_Z^{\top}.
$$

*Proof.* We rewrite the gradient by applying the chain rule for partial derivatives:

$$\nabla_{W_k}\mathcal{L}(t,\mathcal{Y}) = \sum_{i=1}^{n_k} c'_{k(i)} \, \nabla_{W_k} c_{k(i)} + \sum_{i=1}^{n_k}\sum_{j=1}^{q} G'_{k(i,j)} \, \nabla_{W_k} G_{k(i,j)},$$

$$\nabla_{b_k}\mathcal{L}(t,\mathcal{Y}) = \sum_{i=1}^{n_k} c'_{k(i)} \, \nabla_{b_k} c_{k(i)} + \sum_{i=1}^{n_k}\sum_{j=1}^{q} G'_{k(i,j)} \, \nabla_{b_k} G_{k(i,j)},$$

where $G_k \in \mathbb{R}^{n_k \times q}$. Moreover, we have for dimension $i \in [n_k]$ and generator index $j \in [q]$:

$$\nabla_{W_k} c_{k(i)} = \nabla_{W_k}\big(W_{k(i,\cdot)}\, c_{k-1} + b_{k(i)}\big) = e_i\, c_{k-1}^\top, \quad \nabla_{W_k} G_{k(i,j)} = \nabla_{W_k}\big(W_{k(i,\cdot)}\, G_{k-1(\cdot,j)}\big) = e_i\, G_{k-1(\cdot,j)}^\top,$$

$$\nabla_{b_k} c_{k(i)} = \nabla_{b_k}\big(W_{k(i,\cdot)}\, c_{k-1} + b_{k(i)}\big) = e_i, \qquad \nabla_{b_k} G_{k(i,j)} = \nabla_{b_k}\big(W_{k(i,\cdot)}\, G_{k-1(\cdot,j)}\big) = \mathbf{0},$$

where $e_i \in \{0,1\}^{n_k}$ is the $i$-th standard basis vector. Thus,

$$\nabla_{W_k}\mathcal{L}(t,\mathcal{Y}) = c'_k\, c_{k-1}^\top + G'_k\, G_{k-1}^\top = \mathcal{G}_k \odot \mathcal{H}_{k-1}^\top, \qquad \nabla_{b_k}\mathcal{L}(t,\mathcal{Y}) = c'_k = \mathcal{G}_k \odot \langle 1,\mathbf{0}\rangle_Z^\top. \qquad \square$$

**Proposition 15.** *For an input set $\mathcal{H}_{k-1} = \mathcal{Z}$ with $c \in \mathbb{R}^n$ and $G \in \mathbb{R}^{n \times q}$, Alg. 1 has time complexity $\mathcal{O}(n^2\, q)$ w.r.t. the number of input dimensions $n$ and the number of generators $q$.*

*Proof.* Finding the interval bounds of $\mathcal{H}_{k-1}$ (line 2) takes time $\mathcal{O}(n\, q)$ (Prop. 2). Computing the linear approximation (line 4) and the approximation errors (line 5) for each neuron takes constant time; hence, the loop takes time $\mathcal{O}(n)$. The linear map of $\mathcal{H}_{k-1}$ (line 6) takes time $\mathcal{O}(n^2\, q)$ (Prop. 4). Adding the approximation errors (line 7) takes time $\mathcal{O}(n)$. Thus, in total we have $\mathcal{O}(n\, q) + \mathcal{O}(n) + \mathcal{O}(n^2\, q) + \mathcal{O}(n) = \mathcal{O}(n^2\, q)$. $\square$

**Proposition 16.** *The zonotopes during a forward propagation (Prop. 6) have at most $q \le n_0 + \sum_{k \in [\kappa]} n_k$ generators. Let $n_{max} := \max_{k \in [\kappa]} n_k$ be the maximum number of neurons in a layer. The set-based forward propagtion (Prop. 6) has time complexity $\mathcal{O}(n_{max}^2\, q\, \kappa)$ w.r.t. $n_{max}$, $q$ and the number of layers $\kappa$.*

*Proof.* The initial $\epsilon$-perturbance set has $n_0$ generators (line 1) and every nonlinear layer adds $n_k$ new generators for the approximation errors (Alg. 1). Moreover, there are at most $\kappa$ nonlinear layers. Thus, in total, there are at most $(n_0 + \sum_{k \in [\kappa]} n_k)$ generators.

Time Complexity: The $k$-th step of the set-based forward propagation takes time $\mathcal{O}(n_{\max}^2\, q)$: The linear map (line 4) as well as the image enclosure (line 6) takes time $\mathcal{O}(n_{k-1}\, n_k\, q) = \mathcal{O}(n_{\max}^2\, q)$ (Prop. 4 and 15). Thus, the set-based forward propagation (lines 2–6) takes time $\kappa\, \mathcal{O}(n_{\max}^2\, q)$. $\square$

**Proposition 17.** *Alg. 2 has time complexity $\mathcal{O}(n_{max}^2\, q\, \kappa)$ w.r.t. $n_{max}$, $q$ and the number of layers $\kappa$.*

*Proof.* The set-based forward propagation (lines 2–6) takes time $\kappa\, \mathcal{O}(n_{\max}^2\, q)$ (Prop. 16). The gradient of the set-based loss has $(n_\kappa + n_\kappa\, q)$ entries, and the computation of each entry takes constant time; hence, computing the gradient takes time $\mathcal{O}(n_\kappa + n_\kappa\, q)$. The $k$-th step of the set-based backpropagation takes at most $\mathcal{O}(n_{k-1}\, n_k\, q) = \mathcal{O}(n_{\max}^2\, q)$ time: a linear layer computes a linear map (line 11), which takes time $\mathcal{O}(n_{k-1}\, n_k\, q)$ (Prop. 4), and the set-based backpropagation of an image enclosure (line 15) takes time $\mathcal{O}(n_{k-1}\, n_k\, q)$ (Prop. 12). Hence, the set-based backpropagation (lines 9–15) takes time $\kappa\, \mathcal{O}(n_{\max}^2\, q)$. Updating a weight matrix takes time $\mathcal{O}(n_{k-1}\, n_k + n_{k-1}\, n_k\, q) = \mathcal{O}(n_{\max}^2\, q)$ (line 12) and updating a bias vector takes time $\mathcal{O}(n_k) = \mathcal{O}(n_{\max})$ (line 13). There are at most $\kappa$ linear layers; hence, updating the weight matrix and bias vector of all linear layers takes time $\kappa\, (\mathcal{O}(n_{\max}^2\, q) + \mathcal{O}(n_{\max})) = \kappa\, \mathcal{O}(n_{\max}^2\, q)$. Thus, in total, an iteration of set-based training takes time $3\,\kappa\, \mathcal{O}(n_{\max}^2\, q\, \kappa) = \mathcal{O}(n_{\max}^2\, q\, \kappa)$. $\square$

