# OpenReview forum: "Set-Based Training for Neural Network Verification"
_TMLR — Accepted by TMLR_

### Review · Reviewer_5Fqf · 2025-05-03

**Summary Of Contributions:**

The central focus of the paper is the robustness of neural networks: a network is robust if small, carefully chosen input perturbations cannot lead to unexpected output. The paper propose a novel set-based training procedure to achieve robustness with formal verification, where they compute the set of possible outputs given the set of possible inputs and consider a novel gradient set for the entire input. The gradient set controls the size of the output enclosure to improve robustness and simplifies formal verification.

**Audience:**

Yes

**Claims And Evidence:**

Yes

**Requested Changes:**

- The work is more related to Mirman et al. 2018 proposing zonotope enclosure, yet the illustrative Figure 1 does not include any comparison with this baseline.

- The key contribution is utilizing set-based computation instead of single instance-based computation, yet the introduction does not go into details on it. How set-based training increases robustness or simplifies formal verification need to be explained intuitively in the introduction.

- Why is different epsilon chosen for different datasets?

- The experimental result shows a trade-off where the presented method is neither highly accurate nor efficient in verification. Is there a fundamental limitation here that no method can be better in both dimensions?

- The paper mentions polynomial verification of the set-based approach, but I do not find any formal proof in the main paper.

**Strengths And Weaknesses:**

Strengths:

The paper is well-written. Achieving robustness is notoriously hard, specially when one aims to achieve a robust classifier verified formally. I like the set-based approach, however more clarity is needed to showcase the benefit.

Weakness:

Please see below.

---

> ### Author Response · Authors · 2025-05-30
>
> Dear reviewer 5Fqf,
>
> Thank you very much for your time and your valuable comments. We mark our changes in blue in the updated paper for your convenience.
>
> **Illustration of (Mirman et al., 2018) in Fig. 1**
>
> We thank the reviewer for pointing this out. [1] propose a robust training approach using zonotope propagation to bound a maximum loss. However, the zonotope enclosure is only used to compute a tighter enclosure but is boxed again before the attack is computed. Moreover, interval-based approaches perform better [2]. Nevertheless, in the updated paper, we included an illustration for [1] in Fig. 1.
>
> [1] Mirman et al. "Differentiable abstract interpretation for provably robust neural networks." ICML 2018
>
> [2] Jovanović et al. "On the Paradox of Certified Training." TLMR 2022
>
> **Effects of Set-Based Training on Robustness and Verification**
>
> We want to train robust neural networks towards easy subsequent verification.
> Adversarially trained neural networks (e.g., TRADES) achieve great empirical robustness (against adversarial attacks, "falsified Acc." Tab. 1); however, they remain hard to formally verify due to large approximation errors which bloat the size of the computed output set ("fast-verified Acc." Tab. 1). Thus, with our set-based training, we explicitly enforce small output sets through our novel gradient set, which contains a different gradient for each possible output that points towards the target (Fig. 1). The size of the output set cannot be controlled by only using a single gradient, e.g., the gradient corresponding to the maximum loss.
>
> In the updated paper, we clarified this in the introduction (Sec. 1). Moreover, we justify this claim in the evaluation section (Tab. 5).
>
> **Choice of Perturbation Radii**
>
> The perturbation radii for all datasets are taken from previous works to provide comparable results.
> These perturbation radii are also used in the annual neural network verification competition (VNN-COMP).
> To provide further insights, we included a comparison of the maximum perturbation radii for which we can verify a neural network (Appendix B).
>
> **Tradeoff between Accuracy and Robustness**
>
> We agree with the reviewer that an improvement in both robustness and accuracy is desired; however, it has been shown that a fundamental tradeoff exists between the two dimensions [3]. In our approach., we can use the weighting factor in the loss function to tune between accuracy and robustness to strike a good balance. We evaluated the effect of the weighting parameter on the performance in Tab. 2 and clarified the tradeoff in Sec. 5.2.
>
> [3] Zhang et al. "Theoretically Principled Trade-off between Robustness and Accuracy." ICML 2019
>
> **Clarification of Polynomial Verification**
>
> We use a single zonotope propagation through the neural network for the verification, which has polynomial time complexity (Prop. 6). We clarify this in the evaluation section (Sec. 5) of the updated paper. We clarified this in the evaluation section (Sec. 5) of the updated paper.

---

> > ### Comment · Reviewer_5Fqf · 2025-06-27
> > **After rebuttal**
> >
> > I thank the authors for their rebuttal. Most of the concerns are clarified.
> >
> > I am, however, a bit skeptical about the claim of formal verification. Proposition 6 does not directly say anything about polynomial verification complexity. In empirical evaluation, all metrics are related to accuracy. I do not find any run-time results regarding verification, except a line saying `34hours is taken by Gowel et al.".
> >
> > If the claim is to verify robustness formally, some discussion with supporting proofs is needed. I get the point that set-based training, by design, may achieve this, but a formal statement should be made, in my opinion.

---

> > > ### Author Response · Authors · 2025-07-01
> > >
> > > Dear reviewer 5Fqf,
> > >
> > > We agree with the reviewer that this point needs clarification. In the updated paper, we added a proposition for the formal robustness verification of a neural network (Prop. 7) as well as a proposition for the time complexity of a zonotope propagation to verify a neural network (Prop. 15).
> > > In our evaluation, we formally verify all trained neural networks (TRADES, IBP, SABR, and Our's) with a single zonotope propagation for each test input (Prop. 7), without any branch-and-bound procedure.
> > > The verification of a 6-layer convolutional neural network trained on MNIST with 10000 test inputs takes approximately 30min compared to 34h by (Müller et al., 2023, App. C). We clarify this in the evaluation section (Sec. 5.1).

---

### Review · Reviewer_URdj · 2025-05-08

**Summary Of Contributions:**

The paper presents a novel set-based training approach to enhance the robustness of neural networks and facilitate their formal verification, particularly for safety-critical applications. By leveraging zonotopes to represent input and output sets, the authors introduce a gradient set that directly minimizes output enclosures, improving robustness against adversarial perturbations. A set-based loss function balances accuracy and robustness, supported by efficient, differentiable set propagation algorithms. The method is evaluated on datasets such as MNIST, CIFAR-10, SVHN, and TinyImageNet, demonstrating competitive performance and enabling polynomial-time verification.

**Audience:**

Yes

**Broader Impact Concerns:**

No need for this paper to add the broader impact statement.

**Claims And Evidence:**

Yes

**Requested Changes:**

Please address my concerns

**Strengths And Weaknesses:**

First, at the beginning of this submission, the authors tell us that this work is motivated by the adoption of neural networks in safety-critical environments, citing examples like autonomous vehicle control and airborne collision avoidance. The evaluation and methodology focus primarily on image classification tasks (MNIST, CIFAR-10, SVHN, TinyImageNet), which do not fully represent the complexity of safety-critical applications. Thus, the conducted experiments cannot support the claim in the introduction.

Second,  the work is driven by the need to ensure neural network robustness against small input perturbations, particularly in safety-critical environments where noisy sensor data or adversarial attacks are concerns. However, the approach addresses robustness against limited perturbations categories, which is a specific and simplified model of adversarial attacks. If I understand it correctly, such a consideration may not adequately satisfy the requirement in the real world.

Then, the proposed training procedure is computationally intensive compared to other robust training methods. Table VII in the paper shows that the proposed method has a training time of 61.2 seconds per epoch on the MNIST dataset with the CNN6 architecture, significantly higher than baselines like IBP (19.9 seconds), TRADES (49.6 seconds), and point-based training (6.1 seconds). The authors need to clarify whether such a higher computational cost undermines or affects the use of such a framework.

If I understand it correctly, this work relies on dataset-specific hyperparameters for balancing accuracy and robustness. Such reliance on dataset-specific hyperparameters for balancing accuracy and robustness may complicate practical adoption and generalization to new tasks.

---

> ### Author Response · Authors · 2025-05-30
>
> Dear reviewer URdj,
>
> Thank you very much for your time and your valuable comments. We mark our changes in blue in the updated paper for your convenience.
>
> **Evaluation on Classification Benchmarks**
>
> We thank the reviewer for pointing this out. Our paper focuses on training and verifying the robustness of neural networks, which is a necessary property for neural networks in safety-critical environments. We follow previous works in our evaluation and choose standard benchmark datasets for robust neural network training. The extension of our set-based training to train safe and robust neural network controllers for safety-critical environments apart from classification (e.g., in reinforcement learning) is part of our ongoing research.
>
> In our updated paper, we clarified the claim in the introduction (Sec. 1).
>
> **Real-World Perturbations**
>
> This is an excellent comment. We represent all possible perturbations within a $\ell_\infty$-ball with a continuous set and enclose all possible outputs of the neural network.
> We use the enclosure to formally verify that no adversarial attack can exist within a given perturbation radius. However, we do not provide guarantees for attacks outside the perturbation radius.
>
> In our updated paper, we clarified the threat model in the introduction (Sec. 1).
>
> **Training Oveahead**
>
> The reviewer is correct in their assessment that set-based training is computationally more expensive than TRADES, IBP, and SABR. The adversarially trained neural networks (TRADES) have great empirical performance (clean and falsified accuracy Tab. 1) but cannot be verified using polynomial time verification approaches.
> IBP and SABR have significantly lower empirical performance compared to TRADES.
> Our set-based trained neural networks have higher empirical performance than IBP and SABR and have great verified robustness (Tab. 1, fast-verified accuracy).
> Moreover, with our set-based training, we can use a weighting parameter in the set-based loss to explicitly tune the tradeoff between empirical performance and verified robustness.
> Therefore, set-based training balances training time, empirical performance, and verified robustness.
>
> We clarified this in the updated paper (Appendix A).
>
> **Training Hyperparameters**
>
> We agree with the reviewer that relying on hyperparameters can complicate the practical adoption, and automatically tuning this parameter is part of our ongoing research. However, our approach only has a single weighting factor, whereas SABR has four hyperparameters and TRADES has three hyperparameters. Thus, our approach already marks a significant step in this direction.

---

### Review · Reviewer_4XbV · 2025-05-20

**Summary Of Contributions:**

The paper proposes a set-based training approach for promoting verifiable adversarial robustness in neural networks. A novel technique for propagating zonotopes through neural networks is provided, enabled by the introduction of a novel loss function. An experimental comparison with other formal methods approaches is provided, and some ablations are also conducted.

**Audience:**

Yes

**Broader Impact Concerns:**

No broader impact concerns.

**Claims And Evidence:**

No

**Requested Changes:**

A better experimental validation of the benefits w.r.t. Singh et al. (2018) is necessary. The purported mechanism by which the proposed method improves on existing approaches is by finding smaller enclosures, but this is not sufficiently demonstrated. The other suggestions for additional experiments should be considered optional.

Some more discussion on the relative merits of the proposed loss versus existing losses would be appreciated.

**Strengths And Weaknesses:**

The paper is well-presented and quite easy to follow, considering the quite technical nature of the content. The authors have done an above-average job in this respect.

In my view, the main contribution of this paper is a novel approach for bounding the approximation errors of the linearised activation function, resulting in better zonotope propagation through non-linear layers. The benefits of this approach over the well-known method of Singh et al. (2018) are demonstrated quite convincingly at an intuitive level, although the benefits seem to be limited in the experimental setup considered by the authors.

While the set-based training is also interesting, I think there is a lack of meaningful justification for the proposed loss, beyond how it enables one to do set-based training. It would be nice to see it tied to the usual justifications used for motivating a particular loss function (e.g., cross entropy corresponds to maximum likelihood estimation under categorical likelihood, hinge loss upper bounds zero-one loss, etc). From what I can tell the typical approach of taking the maximum loss inside the zonotope will also cause the F-radius of the zonotope to decrease, and it also provides an upper bound to the adversarial loss. Is there is a reason why integrating over the whole zonotope would be more useful in practice? Since it computes an average instead of a maximum, could it result in less robust models?

There are a few other places where I think the experiments could be improved. In particular:
* I would have liked to see how the method scales compared to other approaches as the networks are made deeper---is there a point where the errors introduced by the enclosures compound to the point where this approach is no longer viable? If so, is that point substantially later than when the bound of Singh et al. (2018) is used?
* Ablations are performed on MNIST, a dataset where linear models can already achieve quite high accuracy. In this case, I suspect the overapproximation discussed in Sec 4.1 is not a big problem. Is there a more noticeably difference in performance when more difficult datasets are used for ablations?
* It is generally good practice to investigate how sensitive adversarial defences are to the exact choice of $\epsilon$.
* Adversarial training is very effective in practice, but lacks guarantees; if adversarial training is used to train the network, can the proposed approach provide meaningful guarantees?

---

> ### Author Response · Authors · 2025-05-30
>
> Dear reviewer 4XbV,
>
> Thank you very much for your time and your valuable comments. We mark our changes in blue in the updated paper for your convenience.
>
> **Enclosure of Activation Layers (Our's vs. Singh's): Contribution and Scalability**
>
> We thank the reviewer for highlighting the contribution of our enclosure of activation layers.
> In the updated paper, we included a comparison of our enclosure and Singh's enclosure (Tab. 4), clearly demonstrating that our enclosure results in higher accuracies than Singh's enclosure. Moreover, we are running experiments with larger and deeper neural networks, and we will update the paper once the experiments are finished.
>
> **Justification for the Set-Based Loss Function**
>
> This is an excellent question. We want to train a robust neural network to ease subsequent verification.
> Adversarially trained neural networks (e.g., TRADES) achieve great empirical robustness ("falsified Acc." Tab. 1); however, they remain hard to verify due to large approximation errors which increase the size of enclosure of the output set ("fast-verified Acc." Tab. 1). Thus, with our set-based training we explicitly enforce small output sets through our novel gradient set, which contains a different gradient for each possible output that points towards the target (Fig. 1). The size of the output set cannot be controlled by only using a single gradient, e.g., the gradient of the maximum loss. Further, we do not average over the gradient set; we use the entire gradient set for backpropagation and compute the analytical gradient of the set-based loss function with respect to each neural network parameter. Our set-based loss generalizes the tradeoff loss [1], which splits the robust classification loss into a natural loss and a boundary loss.
> To further illustrate this, we compare the sizes of the output sets (for the first 1000 test samples) and the Lipschitz constant of the best-performing neural network trained on MNIST:
>
> | Method | Size (Interval-Norm) | Lipschitz Constant |
> | - | - | - |
> | TRADES | 276.36 ± 120.40 | **9.72e+07** |
> | IBP | 4.77 ± 1.96 | 2.78e+16 |
> | SABR | 5.26 ± 2.39 | 4.20e+17 |
> | Set (ours) | **1.85** ± 1.27 | 3.49e+12 |
>
> The Lipschitz constant of a neural network is a metric for robustness because it bounds the sensitivity for input changes [2]. The Lipschitz constant of the TRADES-trained network is the smallest, which explains the great empirical robustness; however, the output sets are too large for verification. In contrast, our set-based trained neural network produces the smallest output sets and is verifiable despite a larger Lipschitz constant.
>
> In our updated paper, we clarified the justification of our set-based loss in the introduction (Sec. 1) and compared the interval norm and Lipschitz constant in the evaluation (Sec. 5).
>
> [1] Zhang et al. "Theoretically Principled Trade-off between Robustness and Accuracy." ICML 2019
>
> [2] Fazlyab et al. "Efficient and Accurate Estimation of Lipschitz Constants for Deep Neural." NeurIPS 2019
>
> **Ablation Studies on MNIST**
>
> We agree with the reviewer that linear models can achieve high (clean) accuracy on MNIST.
> However, our primary goal is to train robust neural networks under input perturbation.
> As shown in the table below, the accuracy of linear models quickly deteriorates under input perturbation,
> whereas our set-based models remain at high accuracy.
>
> | Method | clean Acc. | falsified Acc. | fast-verified Acc. (max) |
> | - | - | - | - |
> | Linear Model | 97.46 ± 0.15 | 4.67 ± 0.19 | 0.00 ± 0.00 (0.00) |
> | Set (ours) | **98.76** ± 0.29 | **97.52** ± 0.26 | **95.89** ± 0.82 (**96.40**) |
>
> To provide further insights into how our methods perform on more complex datasets, we included an ablation study on the CIFAR10 dataset in our updated paper (Appendix B, Tab. 9).
>
> **Robustness for fixed $\epsilon$**
>
> We follow previous works and use standard perturbation radii for each dataset. However, it is interesting to compare the maximum perturbation radius. We can use binary search to compute the maximum perturbation radius of the best-performing neural networks on MNIST for the first 1000 test samples:
>
> | Method | $\epsilon$ (max) |
> | - | - |
> | TRADES | 0.0715 ± 0.0140 (0.1094) |
> | IBP | **0.1856** ± 0.0528 (0.2441) |
> | SABR | 0.1792 ± 0.0482 (0.2422) |
> | Set (ours) | 0.1782 ± 0.0494 (**0.2676**) |
>
> The perturbation radii of IBP, SABR, and Set are comparable and significantly higher than those of TRADES. We added this comparison to the updated paper (Appendix B, Tab. 8).
>
> **Verification of Adversarial Trained Neural Network**
>
> Yes, we can apply our formal verification approach to prove the robustness of the adversarially trained neural networks. However, in most cases, the adversarially trained neural networks have large approximation errors, making the verification very hard and time-consuming; e.g., the TRADES networks achieve low verified accuracy (Tab. 1). In contrast, our set-based networks can be formally verified.

---

> > ### Author Response · Authors · 2025-06-02
> >
> > We updated the paper with a comparison of the enclosures using the 6-layer neural network trained on MNIST (Appendix B Tab. 10).

---

### Author Response · Authors · 2025-07-21
**Rebuttal Summary**

Dear Action Editor, Dear Reviewers,

Thank you very much for organizing the review and the valuable feedback. Below, we briefly summarize our main changes.

- We justify our our set-based loss with its ability to explicitly reduce the size of the output sets of neural networks. We showcase that our set-based trained neural networks produce the smallest output sets (Tab. 5).
- We added multiple additional comparisons and ablation studies (Tab. 4, 9, & 10) to provide further insights into how our set-based training performs.
- We clarified some points that reviewers flagged as unclear to improve the readability of our paper.

Best regards, The Authors.

---

### Decision · Action_Editor_v98g · 2025-07-30

**Recommendation:** Accept as is

**Audience:**

Yes

**Audience Explanation:**

The robustness is an active topic and there is lot of interest in techniques that can improve robustness of neural networks. The simplicity of proposed method makes it likely that it would be investigated further by the community.

**Claims And Evidence:**

Yes

**Claims Explanation:**

This paper focuses on how to improve robustness of the neural networks. The proposed insight is amazingly simple: train it with a set of outputs and set of inputs, compute gradient set. The approach is shown to yield networks with good robustness as well as being easy to formally verify.

The reviewers appreciated the contribution of the paper. I encourage authors to give serious consideration to all the comments by the reviewers.